# Interventions promoting occupational balance in adults: A systematic literature review

Stefanie Lentner[1]*, Evelyn Haberl[1], Larisa Baciu[1], Mona Dür[2,3], Cornelia Lischka[2], Mandana Fallahpour[3], Susanne Guidetti[3], Hanna Köttl[1]

1 Department of Health Sciences, IMC University of Applied Sciences Krems, Krems an der Donau, Austria, 2 Duervation GmbH, Krems, Austria, 3 Department of Neurobiology, Care Science and Society, Division of Occupational Therapy, Karolinska Institutet, Huddinge, Sweden

* stefanie.lentner@imc.ac.at

## Abstract

### Introduction

Occupational balance, the subjective perception of satisfaction and balance in engaging in meaningful activities, is fundamental to individuals' health and well-being. The detrimental impacts of decreased occupational balance are increasingly acknowledged, and interventions are emerging. A comprehensive review of these interventions, targeting occupational balance in adult populations, is needed to ensure effective implementation into both clinical and public health settings.

### Objective

This study aimed to systematically review and synthesize existing interventions that address occupational balance among adults in diverse contexts, and to evaluate their effectiveness.

### Method

A systematic literature search was conducted in PubMed, CINAHL, the Cochrane Library, and EMBASE in April 2024, following the PRISMA guidelines. Peer-reviewed articles published between 2000 and 2024, reporting quantitatively on interventions addressing occupational balance, were included. The NHLBI quality assessment tools were employed to evaluate the risk of bias. A narrative synthesis was performed.

### Results

Of the 347 records identified, 18 publications were included in this review. Study designs comprised randomized controlled trials, observational studies, and pre-post studies. Most participants had specific diagnoses, with a predominance of mental health conditions. The review identified 12 interventions aimed at promoting occupational balance, providing an overview of interventions' target groups, goals, features,

**Data availability statement:** All relevant data are within the paper and its Supporting Information files.

**Funding:** This project is funded by the Gesellschaft für Forschungsförderung Niederösterreich m.b.H. (GFF) as part of the RTI-Strategy 2027 (Grant: FTI21-P-005). The funder's website: https://www.gff-noe.at/. GFF had no influence on the research or publication process. This review is part of the CROB project (Collaborative Research on Occupational Balance), which is a research collaboration between the IMC University of Applied Sciences Krems (Austria), Duervation (Austria) and Karolinska Institutet (Sweden).

**Competing interests:** The authors have declared that no competing interests exist.

and content. Overall effectiveness of identified interventions varied across studies, with six demonstrating statistically significant improvements in occupational balance scores. Clinically meaningful changes were observed in areas such as drug craving, social isolation, and work ability.

## Conclusion

This review identified promising interventions for promoting occupational balance and enhancing health, well-being, and life satisfaction across various settings. Further research should employ controlled experimental designs to evaluate interventions addressing occupational balance across diverse populations, addressing gender and age differences while assessing effectiveness across delivery modes and settings.

---

## Introduction

In recent years, the pace and complexity of balancing various occupations of everyday life, such as work, family, and leisure time, have intensified in adulthood. Among other factors, the COVID-19 pandemic and the rise of constant connectivity driven by the digital revolution are driving this development forward [1–4]. Occupational balance is a key concept in health sciences, especially in occupational science and occupational therapy, referring to an "individual's perception of having the right amount of occupations and the right variation between occupations" [[5] p.322]. It is therefore understood as the subjectively perceived satisfaction with and balance between the engagement in activities that are rated as meaningful [6]. In this context, "occupation" encompasses any meaningful everyday activity, including paid and unpaid work, school, household chores, leisure, and even rest [7], which people need to, want to or are expected to do [8].

Certain occupations and lifestyles, such as practicing healthy habits, successfully managing daily demands, or fulfilling psychological needs in balance with personal and environmental conditions, are considered beneficial to well-being, health and quality of life. These practices reduce stressful circumstances and meet essential psychological needs [9]. Modern life, however, leads to increased stress and leaves less time to engage in beneficial activities that contribute to general well-being [10]. Both excessive and insufficient engagement in activities can result in decreased occupational balance, a critical psychosocial determinant of health that can either exacerbate or slow the progression of diseases [11]. Various factors can contribute to an imbalance in daily occupations. These include a lack of time to complete desired or necessary tasks, limited possibilities to manage how time is allocated across activities, a mismatch between desired and required activities, and having either too much or too little to do [5]. Also, boredom due to a lack of stimulating occupations or exhaustion from overstimulation may be seen as responses to decreased occupational balance [12]. Occupations can be further restricted due to a lack of time, resources, or awareness that engaging in meaningful occupations is essential for survival, health and well-being [12].

## The associations between health, well-being, and occupational balance

Over the past decade, scholars have established strong evidence emphasizing the association between occupational balance, subjective health, well-being, life satisfaction, and quality of life [13–16]. For instance, Bejerholm and Eklund [15] explored the relationships between occupational engagement, psychiatric symptoms, quality of life, and self-related variables, i.e., mastery, internal and external control, and sense of coherence. They found that high levels of occupational engagement were associated with higher ratings of self-related variables, fewer psychiatric symptoms, and better quality of life, and vice versa. Employing a structural equation modelling approach, Park and colleagues [13] examined the influence of occupational balance on health, quality of life, and other health-related variables in community-dwelling older adults. Their research identified occupational balance as an independent variable, directly or indirectly affecting subjective health, quality of life, and health-related variables.

Although every human may experience decreased occupational balance to some extent throughout their life course [13,17–20], certain populations, such as informal caregivers [21], homeless persons [22], people undergoing life transitions [23] or individuals with specific diagnoses [24–26] seem to be at a greater risk. Life events such as a stroke, or living through a pandemic, have been shown to amplify subjectively perceived decreased occupational balance [27,28]. For example, earlier research has established associations between occupational balance, subjective health, and well-being in parents of preterm infants with a very low birthweight and in parents of children with cerebral palsy [21,29]. Changes in occupational balance and time-use patterns, which potentially affected health and well-being, have also been reported by university students during the COVID-19 pandemic [23]. Other studies have revealed that people with stress and/or mental disorders, such as depression, anxiety, or schizophrenia, often experience a low occupational balance [26,30,31].

## Enhancing occupational balance through targeted interventions

In recent years, scholars have increasingly acknowledged the health-compromising role of restricted occupational balance and have accordingly promoted the design and implementation of interventions addressing this phenomenon, particularly in relation to mental health. A scoping review on the concept of occupational balance [32] identified three interventions aimed at promoting occupational balance. These interventions were conducted in clinical contexts and included a *therapeutic gardening program* for women living with stress-related disorders [33], a *time use intervention* for individuals with mental illness called *Action Over Inertia* [34], as well as an *occupational therapy group* for children [35]. A more recent scoping literature review focusing on general occupational therapy interventions within mental health [36], also presented the group- and activity-based lifestyle program *Balancing Everyday Life (BEL)* for people with mental illness in specialized and community-based psychiatric service [37]. Additionally, the *Redesigning Daily Occupations (ReDO)* intervention, a group-based program, promotes consideration of individual patterns of daily activities and the balance between them [38].

## Gap of knowledge and research aims

Thus far, research on interventions addressing occupational balance has primarily focused on individuals with mental illnesses in a clinical context [15,16,26,30,31]. Recent evidence, however, indicates that a growing number of studies have explored the health-promoting role of occupational balance in non-clinical contexts [13,14,17,30,39]. To date, no systematic literature review has been conducted to synthesize and assess peer-reviewed publications analyzing interventions that target occupational balance among diverse adult populations and across various disease prevention contexts. These contexts include clinical, community-based, and academic contexts, and encompass populations of different ages, socio-economic situations, diagnoses, and regional backgrounds. Given the significant impact of occupational balance on health, it is crucial to effectively elevate the concept to a public health priority. This approach not only supports the health of the community but is also consistent with societal and economic goals of maintaining a healthy population [9,11].

This systematic literature review therefore aimed to review and synthesize existing interventions that address occupational balance among adults in diverse contexts, as well as to evaluate their effectiveness in promoting occupational

balance. Accordingly, the following research questions were posed: 1) What interventions exist that address occupational balance in adults?, and 2) How effective are existing interventions in promoting adults' occupational balance?

## Method

This systematic literature review followed the Preferred Reporting Items for Systematic Reviews and Meta-analyses (PRISMA) guidelines [40] (S1 Table). The protocol was pre-registered in PROSPERO (#CRD42023423689), an international prospective register for systematic reviews [41]. Amendments to the protocol can be found in S2 Table.

### Selection criteria

Studies were included if they a) had a sample of adults aged 18 years and older, b) involved persons with and without diagnoses, c) reported interventions targeting occupational balance of adults, d) explored occupational balance as primary or secondary outcome, assessed with occupational balance measurement instruments, e) were published between 2000 and 2024, f) used an experimental, quasi-experimental or observational study design and g) were published in a German- or English-language, peer-reviewed journal.

Studies were excluded if they a) were not published in a German- or English-language, peer-reviewed journal (e.g., study protocols, poster presentations), b) the focus was on the occupational balance of persons under 18 years (children and/or adolescents), c) there was no occupational balance measurement instrument being used or d) the study was conducted in a qualitative or mixed-method design.

### Search strategy

The following electronic databases were searched on April 8th, 2024: PubMed, CINAHL, the Cochrane Library and EMBASE. The search string was built by using the PICO framework [42] categories "Person", "Intervention" and "Outcome" and was based on terms and synonyms of ("adult*") AND ("intervention*") AND ("occupational balance"), combining free text words and MeSH terms. A full search strategy is included in the supporting information (S3 Table). The search strategy, incorporating all identified keywords and index terms, was adapted for each included database. The search was re-run prior to the analysis. In addition, a hand-search of reference lists of relevant review articles and included studies was conducted. S4 Table constitutes a numbered table of all studies found in the literature search.

### Study selection

Following the search, all identified citations were collected in Endnote and uploaded to Covidence systematic review management software, which was used to facilitate the screening process [43]. After removing duplicates, two independent reviewers (among SL, EH and HK) conducted the screening of potentially eligible titles and abstracts. Subsequently, full texts were reviewed by two reviewers to determine inclusion in the systematic literature review. A third reviewer was involved in case of disagreement. Cohen's kappa coefficient was calculated for the screening process to evaluate inter-rater agreement and to initiate further discussion in case the coefficient was too low (i.e., *kappa < 0.60*, according to Warrens [44]).

### Assessment of methodological quality

In line with the PRISMA guidelines [40], each study eligible for extraction was critically appraised by two independent reviewers (among SL, EH and HK) using the design-specific standardized National Heart, Lung, and Blood Institute's (NHLBI) quality assessment tools, designed for quality appraisal of study's internal validity [45], which are outlined in S5 Table. In case of disagreement a third reviewer was consulted. To ensure consistency between the reviewers, the tool was pilot tested for each study design, and to assess internal validity, the inter-rater agreement was assessed using Cohen's Kappa [44].

## Data extraction and data analysis

Data extraction was based on the predefined inclusion and exclusion criteria and performed by two reviewers (either SL, EH or HK) individually, using a pilot-tested data extraction form. See S6 Table for all extracted data. The data extraction form included specific details about: author, date, country, conceptualization of occupational balance, study aim, study design, study setting, study participants (inclusion/exclusion criteria, sample size, population description, if applicable: years since diagnosis, method of recruitment, identification of target population, dropouts), type of intervention (goal, duration, format, leader, content and reasoning) and control condition (if applicable), primary and secondary outcome measures, type of statistics, results, conclusions, and study's limitations. For this review, study settings were categorized into clinical, community-based and academic settings. The term "clinical" was used to refer to medical work related to the examination and treatment of individuals based on their health status, including rehabilitation, which involves the process of returning to daily life after illness [46]. The notion "community-based" was defined as a setting that takes place locally, where individuals engage in work, leisure, and other daily activities [47]. "Academic settings" were related to schools, colleges, universities or connected with studying [46]. Additionally, RE-AIM Framework Criteria for conducting literature reviews [48] were considered in the data extraction form as the framework includes dimensions for evaluation of interventions in healthcare and other settings [49–52]. Table 1 presents both the sample size at baseline (N) and the final sample size (n analyzed), which is the number of participants included in the analyses after accounting for dropouts. This ensures that the review is based on the actual number of participants who completed all relevant measurements [62].

A narrative synthesis approach was followed to anayse the data. Narrative synthesis seeks to integrate findings from multiple studies, primarily utilizing descriptive text to summarize and interpret the results, and to draw conclusions based on the body of evidence. It aims to elucidate the mechanisms by which interventions are either effective or ineffective [63]. The main elements of the synthesis were the organization of findings to identify patterns across the studies, the exploration of relationships in the data to explain differences in effects, and the drawing of conclusions about the size and direction of effects.

## Results

The results section provides an overview of the study characteristics and a quality assessment of the included studies. In line with the two research questions, it further comprises a presentation of interventions addressing occupational balance, and an appraisal of their effectiveness. A PRISMA flow diagram [40] is used to summarize the results of the systematic search (Fig 1). A total of 347 publications were identified from databases and hand searches. Duplicates were removed and 256 studies remained. Screening based on title and abstract led to 205 studies being excluded. Reasons were occupational balance only being addressed as a key word, a study population under the age of 18, no evaluation of the effect of the intervention, qualitative or mixed-method design and publication in a non-peer-reviewed journal. Full texts of the remaining 51 studies were assessed for eligibility. Another 33 studies were excluded for not meeting the inclusion criteria: study protocols (n = 5), trial registrations (n = 7), study designs that did not align with the predefined criteria (n = 7), non-peer-reviewed articles (n = 4), or because occupational balance was not addressed (n = 10). This resulted in the inclusion of 18 studies in this systematic literature review. Cohen's kappa coefficient was 0.788 for title and abstract screening and 0.792 for full-text screening. According to Landis and Koch [64], inter-rater reliability can therefore be considered high, and no further discussion of inclusion was necessary.

### Study characteristics

Characteristics of the studies involved are presented in Table 1. The included studies were published between 2011 and 2024 and were conducted in nine countries, with the highest number coming from Sweden (n = 10) [37,39,53–56,65–68]. One study each was carried out in Denmark [60], Ireland [61], Turkey [58], Iran [69], Republic of Korea [57], China [59], Canada [34], and Brazil [70]. All included studies were conducted in English.

**Table 1. Characteristics of included studies.**

| Intervention Author (year) | Study design | Study setting | Study aim | Population under research (diagnosis, mean age, gender, sample size) | Outcome measures | Findings |
|---|---|---|---|---|---|---|
| *Balancing Everyday Life (BEL)* Argentzell et al. (2020) [53] | RCT pre, post intervention and 6 months follow-up | Clinical (outpatient mental health service in Sweden) | - To explore the effects of two activity-based interventions *(BEL, standard occupational therapy)* on personal recovery among service users<br>- to investigate if various aspects of activity may mediate change in recovery while also acknowledging clinical, sociodemographic and well-being factors | Analyses from the same study cohort as in [37]. Persons with a diagnosed mental illness N = 226; n analyzed = 159 (EG: n = 89, CAU: n = 70); dropout rate: baseline to 16 weeks: 20.4% (EG: 24.8%; CG 14%); baseline to 6 months: 26.6% (EG: 33.1%; CG 24.7%) | **Occupational engagement (POES), Satisfaction with Daily Occupations and Occupational Balance (SDO-OB)**, Personal recovery (QPR), Self-Mastery (Pearlin Mastery Scale), Level of Functioning (GAF) | No significant main effect, F (1,214.5) = 0.154, p > 0.05 and no interaction, F (2, 348.6) = 0.151, p > 0.05, was found when comparing *BEL* and standard occupational therapy: there was no difference found regarding recovery improvement. No significant relations between recovery and sex or age. Personal recovery of participants in both groups increased after treatment and further at the follow-up. The strongest mediators for treatment effect were activity engagement (POES) and mastery. |
| *Action Over Inertia (AOI)* Edgelow et al. (2011) [34] | RCT pre and post intervention | Community-based (community treatment services in Canada) | To assess the efficacy and clinical utility of a new occupational time-use intervention | Persons with serious mental illness Age (mean, SD, range): EG: 44.6 (8.38, 31–60), CG: 32.38 (9.40, 21–48); Gender: not recorded; years since diagnosis (mean, SD, range): EG: 21.2 (8.08,11–34), CG: 10.75 (7.59, 3–23); years served by Assertive Community Treatment (mean, SD, range): EG: 6.2 (4.71, 1–17), CG: 2.94 (1.9, 1–7) N = 24; n analyzed = 18 (EG: n = 10; CG: n = 8); dropout rate: 25.0% (EG, 28.6%; CG: 20%) | **Occupational Balance (Time Use Diaries), Occupational engagement (POES)** | OB was measured by time use. Results of the EG showed a shift from sleep to increased general activity (p = 0.05). While EG decreased its time spent in sleep by 47 min/day, CG increased time spent in sleep by 22 min/day at post-test. EG and CG did not differ on the occupational engagement measure. None of the nine categories of the POES showed any significant differences. |
| *Balancing Everyday Life (BEL)* Eklund et al. (2017) [37] | RCT measured pre, post intervention and 6 months follow-up | Clinical (outpatient units and day centre in Sweden) | To evaluate the effectiveness of the 16-week *BEL* program, compared to care as usual (CG) for people with mental illness in specialized and community-based psychiatric services. | Persons with diagnosed mental illness and persons having a self-reported occupational imbalance Age (mean, SD): EG: 40 (11), CG: 40 (11); Gender (% women): EG: 77, CG: 67; self-rated diagnosis is anxiety/bipolar/depressive disorder: EG: 52%, CG: 50% N = 226; n analyzed = 159 (EG: n = 89, CAU: n = 70); dropout rate: from baseline to 16 weeks: 20.4% (EG: 24.8%; CG 14.0%); from baseline to 6 months: 26.6% (EG: 33.1%; CG 24.7%) | **Activity engagement (POES), Satisfaction with Daily Occupations and OB (SDO-OB), Activity Value (Oval-pd)**, Quality of life (MANSA), Self-esteem: Rosenberg self-esteem scale; Self-rated health (SF-36), Psychosocial functioning (GAF) | From pre to post intervention the *BEL* group improved more than CG in some aspects:<br>-highly significant (p < 0.001): increased activity engagement<br>-significant: increased activity level (p = 0.036), more optimal general activity balance (p = 0.042), reduced symptom severity (p = 0.046), increased psychosocial functioning (p = 0.018).<br>-increased general quality of life (p = 0.061) At follow-up the *BEL*-group improved more than CG regarding activity engagement (p = 0.001), activity level (p = 0.007) and general quality of life (p = 0.049). |

*(Continued)*

| Intervention Author (year) | Study design | Study setting | Study aim | Population under research (diagnosis, mean age, gender, sample size) | Outcome measures | Findings |
|---|---|---|---|---|---|---|
| **Balancing Everyday Life (BEL)** Eklund et al. (2023) [54] | RCT pre and post intervention | Community-based (day centre in Sweden) | To compare two groups who received community-based day centre services: -*Balancing Everyday Life (BEL)* intervention -Control group: Standard DC support | The study cohort is part of the study cohort in [37]. Persons with a mental health problem Age (mean, SD): EG 45 (12), CG: 45 (10); Gender (% women): EG: 74, CG: 55; living with a partner: EG 48%, CG: 49%; having children EG: 30%, CG: 11%; self-rated diagnosis is anxiety/bipolar/depressive disorder: EG: 39%, CG: 40% $N=65$; $n$ analyzed = 55 (EG: $n=21$, CG: $n=34$); dropout rate: 15.4% (EG: 22.2%; 10.5%) | Self-developed questionnaire on motivation, **Occupational engagement (POES)**, Personal recovery (QPR), Satisfaction with the DC services | *BEL* participants improved occupational engagement (POES) and personal recovery (QPR) at the end of the intervention, while this was not the case for the control-group over the same period. When comparing the changes in the *BEL* group and the control group, the only difference after the intervention that reached the significance level was occupational engagement (POES), where the *BEL* group improved more (p = 0.004). |
| **Tree Theme Method® (TTM)** Gunnarsson et al. (2018) [55] | RCT pre and post intervention | Clinical (primary health care centres and general outpatient mental healthcare units in Sweden) | To compare *TTM* with regular OT (CG) regarding activities in everyday life, psychological symptoms of depression and anxiety, and health-related and intervention-related aspects before and after the intervention in people with depression and/or anxiety disorders | Persons with depression and/or anxiety disorders Age (mean, SD, range): EG: 43.0 (11.3, 19–63), CG: 40.1 (12.6, 20–64); Gender (% women): EG: 84, CG: 82; affective disorders: EG: 65%, CG: 71%; anxiety disorders: EG: 36%, CG: 29%; living with someone: EG: 69%, CG: 57%; sick leave: EG: 55%, CG: 59% $N=121$; $n$ analyzed = 107 (EG: $n=55$, CG $n=52$); dropout rate: 11.6% (EG: 12.7%; CG: 10.3%) | **COPM, SDO, OBQ**, SCL-90-R, MADRS-S, HADS, SOC, Mastery Scale, MANSA, Haq-II, CSQ | *TTM* and CG both improved from pre to postintervention in all measured outcomes. The changes were statistically significant except for the SDO in the *TTM* group and the Mastery Scale in the CG. No statistically significant differences were found between the groups in any of the measured aspects. |
| **Tree Theme Method® (TTM)** Gunnarsson et al. (2022) [56] | RCT pre, post, 3 months and 12 months follow-up | Clinical (primary health care centres and general outpatient mental healthcare units in Sweden) | To investigate the longitudinal outcomes of the *TTM* compared with care as usual (CG), provided by occupational therapists, in terms of everyday occupations, psychological symptoms, and health-related aspects | Analyses from the same study cohort as in [55] $N=121$; $n$ analyzed = 84 (EG: $n=42$, CG: $n=42$); dropout rate: 30.6% (EG: 33.3%; CG: 27.6%) | **COPM, SDO, OBQ**, SOC, MANSA, SCL-90-R, HADS, MADRS-S | At follow-ups (3 and 12 months): Both groups significantly improved (p-value ≤ 0.01) in everyday occupations, psychological symptoms, and health-related aspects. No significant differences were found between the groups. From baseline to 3 months-follow-up: TTM + CG: significantly improved performance of everyday occupations and their satisfaction with this. Further improved to 12-months-follow-up. *TTM* + CG: statistically significant higher OB (OBQ) at 3 months and 12 months. |

*(Continued)*

| Intervention Author (year) | Study design | Study setting | Study aim | Population under research (diagnosis, mean age, gender, sample size) | Outcome measures | Findings |
|---|---|---|---|---|---|---|
| *time-use intervention* Jung et al. (2023) [57] | RCT measured at admission and discharge | Clinical (community-based hospital in Republic of Korea) | To determine the effectiveness of a *time-use intervention* (EG) on the occupational balance of isolated patients with coronavirus disease | Persons with coronavirus disease Age (mean, SD): EG: 47.37 (17.61), CG: 58.59 (13.51); Gender (% women): EG: 63.16, CG: 45.45; Duration of isolation (days): EG: 13.84, CG: 13.59; Work (% yes): EG: 84, CG: 59 *N*=50; *n* analyzed=41 (EG: *n*=19, CG: *n*=22); dropout rate: 18.0% (EG 24.0%, CG: 12.0%) | **Occupational imbalance (K-LBI)**, depression (PHQ9), anxiety (SAS) Insomnia (ISI-K), boredom (MSBS-8), fear of COVID-19 (FCV-19S), health-related quality of life (WHOQOL-BREF) | EG improved significantly in all measures from admission to discharge, while CG worsened in all aspects. A time×group analysis showed that EG improved all items of OB compared to CG (F=14.12, p<.001). |
| *web-based time-use intervention* Pekçetin et al. (2021) [58] | RCT pre, post intervention | University in Turkey | To evaluate the effectiveness of a *web-based time-use intervention* on the OB of university students | Students Age (mean, SD): EG: 19.66 (0.99), CG: 19.60 (1.32); Gender (% women): EG: 70, CG: 80 *N*=60 (EG: *n*=30; CG: *n*=30); no dropouts | **OB (OBQ-11 T)** | At baseline the total OBQ-11 T score in EG was significantly lower than in CG. From before to after the intervention the EG improved significantly on all but one item. The CG showed significant improvements only in two out of 11 items. |
| *leisure intervention* Farhadian et al. (2024) [67] | Pre-post Study pre, post and 8 weeks follow up | Community-based (OT centre and sports centre in Iran) | To investigate the effect of a *leisure intervention* on occupational performance and occupational balance in individuals with substance use disorder | Persons with substance use disorder (SUD) Age (mean, SD): 30.11 (7.99); Gender (% women): 0; duration of drug use in years (mean, SD): 4.66 (2); abstinence period in months (mean, SD): 21 (10.75) *N*=12; *n* analyzed=9; dropout rate: 25.0% | **Occupational performance and satisfaction (COPM), OB (OBQ11)**, leisure activities (NLQ), health-related quality of life (SF-36), current drug cravings (DDQ) | Significant improvements from pre to post-intervention in occupational performance (COPM-P, p<0.01, r=0.59) and occupational balance (OBQ11, p<0.01, r=0 59). The effect sizes decreased from post-intervention to follow-up, but those differences were not significant. |
| *ballroom dancing* Ferreira de Sousa et al. (2022) [70] | Pre-post Study pre and post intervention | University (in Brazil) | To evaluate the effectiveness of *ballroom dancing* as an occupational therapeutic intervention strategy to reduce stress and promote OB in university students attending courses in the health area | Students from four undergraduate health courses Age range: 19–29; Gender (% women): 55 *N*=18; no dropouts | Stress level (ISSL), QOL (WHOQOL-BREF), **Questionnaire on Occupational Balance** | The stress level changed significantly: at baseline 94.4% had symptoms of stress, post intervention only 27.7% reported symptoms. The participants showed improved results in QOL and OB after the intervention. |
| *occupation-based sleep intervention* Ho et al. (2022) [59] | Pre-post Study pre, post intervention, 1 month and 3 months follow-up | Clinical (general out-patient clinic in China) | To evaluate the effectiveness of an *occupation-based sleep intervention* among community-dwelling adults with insomnia, when compared with a treatment-as-usual group which focused on sleep hygiene, and relaxation | Persons with insomnia disorder Age (mean, SD, range): 57 (5.85, 20–65); Gender (% women): 77; mean onset of sleep problem (mean, SD): 11.64 (5.06) months *N*=46; *n* analyzed=42 (EG: *n*=20, CG: *n*=22); dropout rate: 8.7% (EG: 9,1%; CG: 8,3%) | Insomnia (C-ISI), Sleep quality (C-PSQI), Activity Wristband, **OB in daily life (OB-Quest)**, Personal health (PHQ9), Anxiety (GAD 7) | Comparing of groups: EG had statistically significant higher improvements compared to the CG in the following aspects: Insomnia (F=9.42, p<.001), OB (F=21.74, p<.001), Anxiety (F=7.01, p<.05) and personal health (F=5.76, p<.05). The positive changes of the OB-Quest-results in the EG were higher for the period from before treatment to follow-up (p<.001), than from baseline to post treatment (p<.01). |

*(Continued)*

| Intervention Author (year) | Study design | Study setting | Study aim | Population under research (diagnosis, mean age, gender, sample size) | Outcome measures | Findings |
|---|---|---|---|---|---|---|
| *Let's Get Organized (LGO)* Holmefur et al. (2019) [60] | Pre-post Study pre, post intervention and 3 months follow-up (only ATMS-S) | Clinical (outpatient psychiatric and rehabilitation setting in Sweden) | To pilot test the first part of the *Let's Get Organized (LGO)* occupational therapy intervention in a Swedish context by exploring enhancements of time management skills, aspects of executive functioning, and satisfaction with daily occupations in people with time management difficulties because of neurodevelopmental or mental disorders | Persons with confirmed/suspected mental or neurodevelopmental disorder Age (mean, SD, range): 34.3 (10.1, 20–62); Gender (% women): 69; Working: 38%; Unemployed: 62% N=75; n analyzed=55; dropout rate: 26.7%; further attrition to 3-months follow up: 40.0% | Time management skills (ATMS-S), Weekly Calendar Planning Activity (WCPA–SE), **Satisfaction with daily occupations (SDO-13)** | From pre intervention to post intervention the scores for all subscales of the ATMS-S improved significantly (p<0.001). The results further improved to 3-months-follow-up, with one of the items (organization and planning) reaching the level of significance. The evaluation of the SDO-13 shows that the number of activities carried out increased significantly from 7.3 before to 8.3 after the intervention (p=0.001). During this period the mean satisfaction with the activities increased from 54.9 to 61.1 (p<0.001) and the mean overall satisfaction improved from 3.2 to 2.8 (p=.007; lower values indicate greater satisfaction). |
| *ReDesign your EVEryday Activities and Lifestyle with Occupational Therapy - REVEAL(OT)* Nielsen et al. (2024) [60] | Pre-post Study pre and post intervention | Clinical (multi-disciplinary pain centre in Denmark) | To investigate various pre-post changes in adults with chronic pain participating in *ReDesign your EVEryday Activities and Lifestyle with Occupational Therapy - REVEAL(OT)* | Persons with chronic non-malignant pain Age (mean, SD): 46.6 (10.9); 80% were 35 or older; Gender (% women): 85; did not enter the higher education system: 74.1%; received social-supportive economic benefits: 77.8%; experienced more than 50% of the body regions affected by pain: 70.4%; average years of pain experience: 11; average pain intensity score (0–10 scale): 6.5 N=40; n analyzed=31; dropout rate: 22.5%; 1 participant excluded due to not meeting the inclusion criteria | Pain catastrophizing (PCS), Pain intensity (BPI-sf), Health (EQ-5D-5L), Pain spreading: body pain chart, Pain acceptance (CPAQ), Pain self-efficacy (PSEQ), **OB (OBQ)**, Sleep Quality: (KSQ), Motor and process skills (AMPS), BMI, blood pressure, waist-circumference, pain sensitivity, physical wake-time activity (PWTA) | Statistically significant improvements in motor skills (AMPS: mean difference 0.20 (1.37;1.57), 95% CI 0.01; 0.38) and temporal summation of pain (−1.19 (2.86; −1.67), 95% CI −2.16; −0.22), decrease in pain tolerance (cPTT right leg mean difference: −7.110 (54.42; 47.32), 95% CI −13.99; −0.22) No statistically significant pre-post changes in self-perceived health status, process skills, occupational balance. Significant correlation between improved pain self-efficacy (PSEQ) and occupational balance (OBQ) (r=0.58). High positive (but not statistically significant) correlation between OBQ and COPM satisfaction with occupational performance (r=0.52). |

*(Continued)*

| Intervention Author (year) | Study design | Study setting | Study aim | Population under research (diagnosis, mean age, gender, sample size) | Outcome measures | Findings |
|---|---|---|---|---|---|---|
| *psycho-educational program* Ryan et al. (2023) [61] | Pre-post Study pre and post intervention | Clinical (inpatient addiction recovery service, private health care in Ireland) | To explore the impact of an occupational therapy-led intervention on self-reported occupational performance and OB issues for people living with SUDs (substance use disorders) within an inpatient addiction service | Persons with substance use disorder Age range: 18–65 +; Age distribution: [45–52,62,63]: 31.3%; [35–44]: 25%; [53–56,60,64–68]: 18.8%; Gender (% women): 43,7; main healthcare concern: substance misuse (68.8%) N = 21; n analyzed = 15; drop-out rate: 28.6%; 1 participant excluded due to incomplete data | **Recovery (C-PROM)** | 13 out of 15 participants increased their C-PROM-score from pre to postintervention. the mean score of the group was significantly higher after the intervention (p = 0.007). The majority of the OB-related questions showed more positive results after the intervention. |
| *Tree Theme Method® (TTM)* Hakansson et al. (2023) [39] | Cohort study post intervention and 12 months follow-up | Clinical (primary health care and outpatient mental healthcare units in Sweden) | - To explore associations between different aspects of OB and satisfaction with daily occupations - to explore whether different aspects of OB predicted satisfaction with daily occupations 12 months later. | Analyses from the same study cohort as in [55], specific data collection points: directly after the treatment, and 12 months later N = 121; n analyzed = 84; drop-out rate: 30,6% | **OB (OBQ11), Satisfaction with daily occupations (SDO-13)** | Post intervention statistically significant associations were found between: Balance between work, home, family, leisure, rest, and sleep; Having neither too much nor too little to do during a regular week; Satisfaction with time spent in rest, recovery, and sleep and satisfaction with daily occupations. The item Balance between energy- giving/energy-taking occupations at post-intervention predicted satisfaction with daily occupations at follow-up. |
| *Balancing Everyday Life (BEL)* Hultqvist et al. (2019) [66] | Cohort study pre, post intervention and 6 months follow-up | Clinical (outpatient mental health service in Sweden) | To explore which baseline factors could predict clinically important improvements in occupational engagement, activity level, occupational balance and QoL among mental health service users at *BEL* completion and to follow-up | Persons with diagnosed mental illness and having a self-reported occupational imbalance The study cohort is part of the study cohort in [37]. Age (mean, SD): 40 (11); Gender (% women): 77; living with a partner: 30%; having children: 47%; Living situation: Own house or flat, no support 66%; Own house or flat, with support 25%; supported housing 2%; Lodging 7% N = 133; n analyzed = 89; drop-out rate: 33,1% | Psychosocial functioning (GAF), Self-factors (Rosenberg self-esteem scale, Pearlin Mastery Scale), **Activity engagement (POES), Satisfaction with Daily Occupations and Occupational Balance (SDO-OB)**, Quality of Life (MANSA) | Several of the explored baseline factors (care context and socio-demographic, clinical and self-related factors) were associated with clinically important improvements. However, the multivariate analyses identified only a few predictors: Having a close friend predicted improving in the leisure domain of OB. Female gender predicted increasing the results for the self-care domain, and self-esteem for the home chores domain. |

*(Continued)*

| Intervention Author (year) | Study design | Study setting | Study aim | Population under research (diagnosis, mean age, gender, sample size) | Outcome measures | Findings |
|---|---|---|---|---|---|---|
| ***ReDesigning Daily Occupation Program (ReDO-10)*** Olsson et al. (2020) [67] | Longitudinal cohort study pre, post intervention and 12 months follow-up | Clinical (primary health care centers in Sweden) | To investigate if the occupation-based intervention *ReDesigning Daily Occupation Program (ReDO-10)* predicts work ability in long-term perspective for women at risk for or on sick-leave. | Women on or at risk for sick leave Age (mean): 56; Age distribution: <30: 6%; [30–39]: 12%; [40–49]: 45%; [50–56,62–64]: 29%; >60: 8%; Gender (% women): 100%; education: elementary school 6%; professional school 31%; 3–4 years upper secondary high school 35%; university 28% $N$=152; $n$ analyzed=86; drop-out rate: 43,4% | Work ability (WAI), **OB (OBQ), Occupational Value (OVal-pd)**, Mastery (Mastery-S), Perceived Health (EQ-VAS) | Statistically significant improvements from before to after the intervention for OB (p<0.001), mastery (p<0.001), occupational value (p<0.001), perceived health (p<0.001) and work ability (WAI single item, p<0.001). From before the intervention to follow-up there were significant increases for occupational balance (p=0.002), perceived health (p=0.005) and work ability (WAI single item, p=0.003) but not for mastery (p=0.555) or occupational value (p=0.715). |
| ***Tree Theme Method® (TTM)*** Wagman et al. (2023) [68] | Cohort study pre, post intervention, 3 months and 12 months follow-up | Clinical (primary health care and outpatient mental healthcare units in Sweden) | To describe and to compare the self-rated quality of life (QOL), sense of coherence and OB after participation in OT in three groups of people based in their work situation during the study period: continuous sick leave (SL), returned to work (RTW) and continuous work (W) | The study cohort is part of the study cohort in [55]. Age (mean, range): SL: 51 (21–60), RTW: 42 (19–60), W: 39 (20–57); Gender (% women): SL: 92, RTW: 85, W: 82; having children under 18 years: SL: 25%, RTW: 23%, W: 18%; having friends: SL: 88%, RTW: 92%, W: 88% $N$=54; complete data sets used | QOL (MANSA), sense of coherence (SOC), **OB (OBQ)** | No significant difference in QOL, sense of coherence or OB between the groups on any occasion. Changes over time, by groups: RTW and W: QOL increased significantly from pre to post intervention and even further et the follow-ups; W: SOC increased significantly from pre to post intervention and to follow-up after 3 months; RTW and SL: OB increased significantly between the individual data collections. |

*Note*. Studies are listed by study design and alphabetically; Outcome measures on occupational balance are highlighted bold. Additional information on the interventions is reported in Table 3. *Abbreviations*. CG: control group; EG: experimental group; OB: occupational balance; OT: Occupational Therapy; QOL: quality of life. Outcome measures: AMPS: standarized Assessment of Motor and Process Skills; ATM-S: Swedish Version of the Assessment of Time Management Skills; BMI: Body Mass Index; BPI-sf: Brief Pain Inventory Short Form; C-ISI: Cantonese Version Insomnia Severity Index; **COPM: Canadian Occupational Performance Measure; COPM-P: Canadian Occupational Performance Measure – Performance; COPM-S: Canadian Occupational Performance Measure – Satisfaction;** CPAQ: Chronic Pain Acceptance Questionnaire; **C-PROM: Canadian Personal Recovery Outcome Measure;** C-PSQI: Chinese Version Pittsburgh Sleep Quality Index; CSQ: Client Satisfaction Questionnaire; DDQ: Desire to Drug Questionnaire; EQ-5D-5L: EuroQoL questionnaire; EQ-VAS: EuroQol-visual analog scale; FCV-19S: Fear of COVID-19 scale; GAD7: General Anxiety Disorder 7; GAF: Pearlin Mastery Scale Global Assessment of Functioning; HADS: Hospital Anxiety and Depression Scale; HAq-II: Helping Alliance questionnaire; ISI-K: Korean version of Insomnia Severity Index; ISSL: LIPP's Inventory of Stress Symptoms for Adults; **K-LBI: Korean version of the Life Balance Inventory;** KSQ: Karolinska Sleep Questionnaire; MADRS-S: Montgomery-Asberg Depression Rating Scale; MANSA: Manchester Short Assessment of Quality of Life; Mastery-S: Perlin Master Scale; MSBS-8: Multidimensional State Boredom Scale-8; NLQ: Nottingham Leisure Questionnaire; **OBQ: Occupational Balance Questionnaire; OBQ-11: revised on 11 (of 13) items; OB-Quest: Occupational Balance Questionnaire; OVal-pd: Occupational Value with predefined items;** PHQ9: patient Health Questionnaire-9; **POES: Profiles of Occupational Engagement in people with Severe mental illness;** PSEQ: Self Efficacy Questionnaire; QPR: Questionnaire about the Process of Recovery; SAS: Zung's Self-rating Anxiety Scale; SCL-90R: Symptom Checklist-90-R; **SDO-OB: self-report version Satisfaction with Daily Occupations and Occupational Balance;** SF-36: 36-Item Short-Form Health Survey; SOC: Sense of Coherence scale; WAI: Work Ability Index; WCPA–SE: Weekly Calendar Planning Activity; WHOQOL-BREF: World Health Organization Quality of Life Assessment Instrument.

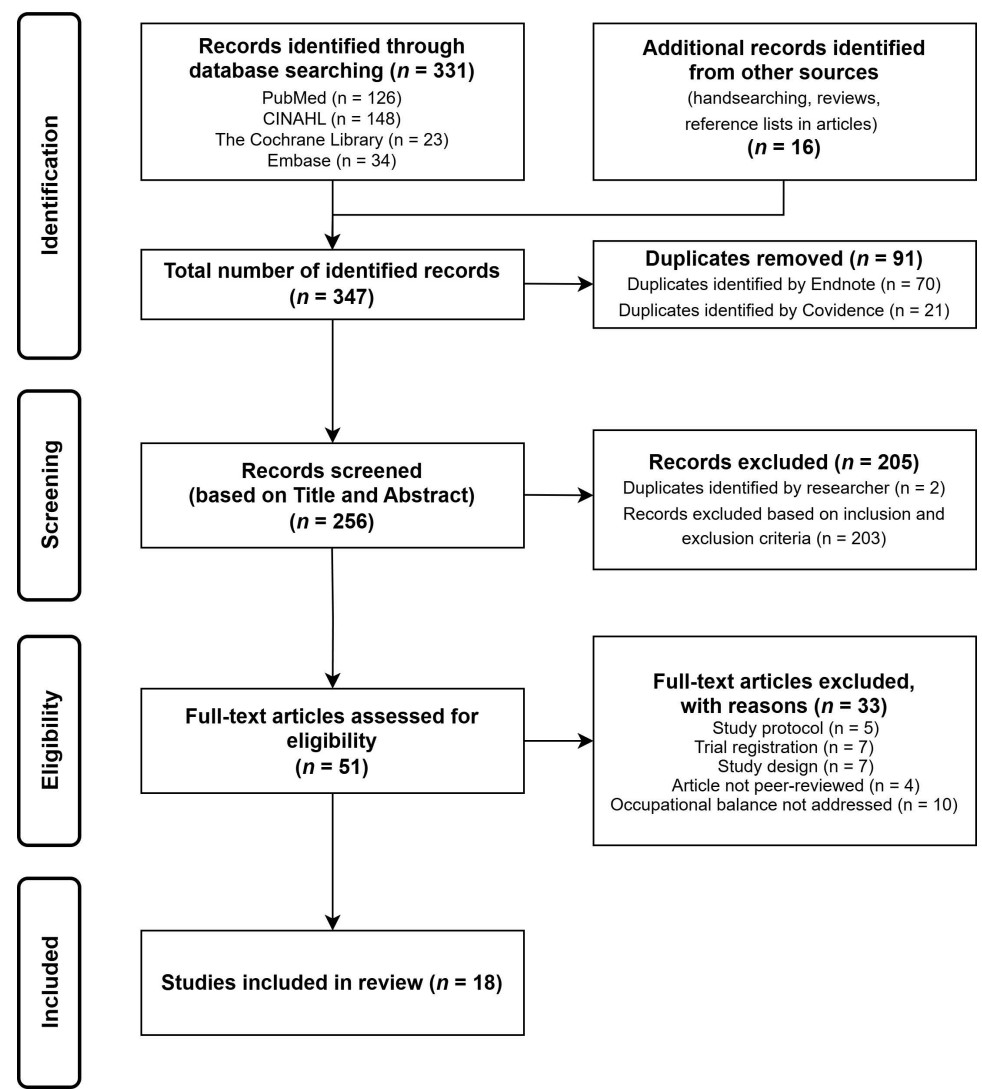

**Fig 1. Prisma flow chart of the literature search and review process.**

The study designs of the included publications comprised randomized controlled trials (RCT) (*n*=8) [34,37,53–56,58,57], pre-post (*n*=6) [65,60,61,69,59,70] and observational (*n*=4) [39,66–68] studies. The sample size at baseline (*N*) varied between 12 [69] and 226 [37] participants across all studies, involving diverse populations, which are described in detail below. A total of 641 study participants completed all study measurements and were included in the analyses at the study endpoints (i.e., *n* analyzed). Fourteen studies included more women than men, while one study had a higher proportion of male participants [61]. One study targeted only women [67], another focused exclusively men [69], and one study did not record participants' gender [34]. Participants' mean age ranged from 19 to 56 years. Commonly reported demographic data encompassed years since diagnosis (if applicable), health status, educational level, and living situation. Research was conducted across a variety of settings, including primary health care centers, outpatient units, hospitals, inpatient addiction recovery, therapy centers, day services, sports centers, and universities. The settings were categorized into clinical (*n*=13) [37,39,53,55,56,60,61,65–68,57,59], community-based (*n*=3) [34,54,69] and academic (*n*=2) [58,70]. Additional information on the interventions is presented in Table 3.

## Assessment of methodological quality

The appraisal of study quality [45] based on 12 or 14 criteria is presented in Table 2. The critical appraisal questions (14 questions for RCTs and observational studies, and 12 questions for pre-post studies) are provided in the supporting information (S5 Table). Cohen's kappa coefficient was 0.795, indicating substantial inter-rater agreement [44]. The overall study quality of included references was deemed as good in five studies [37,55,56,58,59], moderate in eleven studies [39,53,54,60,61,65–68,69,57] and poor in two studies [34,70].

**Table 2. Quality appraisal of included studies (NHLBI quality assessment tool).** [53,34,37,54,55,56,57,58,69,70,59,65,60,61,39,66,67,68].

| Author (year) | X.1 | X.2 | X.3 | X.4 | X.5 | X.6 | X.7 | X.8 | X.9 | X.10 | X.11 | X.12 | X.13 | X.14 | yes |
|---|---|---|---|---|---|---|---|---|---|---|---|---|---|---|---|
| **RCTs** | | | | | | | | | | | | | | | |
| Argentzell et al. (2020) [56] | + | + | + | - | + | + | - | + | + | / | + | - | + | + | 71% |
| Edgelow et al. (2011) [34] | + | / | / | / | / | - | - | + | + | - | + | - | + | - | 36% |
| Eklund et al. (2017) [37] | + | + | + | / | + | + | - | + | + | + | + | + | + | + | 86% |
| Eklund et al. (2023) [57] | + | + | / | / | / | - | + | + | + | / | + | - | + | - | 50% |
| Gunnarsson et al. (2018) [58] | + | + | + | + | + | + | + | + | + | / | + | + | + | + | 93% |
| Gunnarsson et al. (2022) [59] | + | + | + | / | + | + | - | + | + | + | + | - | + | + | 79% |
| Jung et al. (2023) [68] | + | + | / | - | + | - | + | + | + | + | + | - | + | + | 71% |
| Pekçetin et al. (2021) [66] | + | + | + | - | + | + | + | + | + | / | + | + | + | + | 86% |
| **Pre-Post-Studies** | | | | | | | | | | | | | | | |
| Farhadian et al. (2024) [67] | + | + | - | / | - | + | + | / | - | + | - | + | n/a | n/a | 50% |
| Ferreira de Sousa et al. (2022) [70] | + | - | - | - | - | / | - | / | / | + | - | - | n/a | n/a | 17% |
| Ho et al. (2022) [69] | + | + | / | + | + | + | + | + | + | + | - | - | n/a | n/a | 75% |
| Holmefur et al. (2019) [60] | + | + | + | / | / | + | + | - | - | + | - | - | n/a | n/a | 50% |
| Nielsen et al. (2024) [64] | + | + | - | / | - | + | + | / | + | + | - | + | n/a | n/a | 58% |
| Ryan et al. (2023) [65] | + | + | + | + | - | + | + | - | - | + | - | + | n/a | n/a | 67% |
| **Observational Cohort and Cross-Sectional Studies** | | | | | | | | | | | | | | | |
| Håkansson et al. (2023) [39] | + | + | / | + | - | + | + | / | / | / | + | + | - | + | 57% |
| Hultqvist et al. (2019) [61] | + | + | / | + | + | + | + | / | + | + | + | - | - | + | 71% |
| Olsson et al. (2020) [62] | + | + | + | + | - | + | + | / | + | / | + | / | - | + | 64% |
| Wagman et al. (2023) [63] | + | + | + | + | / | / | + | / | + | / | + | / | - | - | 50% |

*Note.* +, yes; -, no; /, cannot determine or not reported; n/a, not applicable. Studies are grouped according to study design and sorted alphabetically. Study quality is indicated as percentages of "yes" answers in the right-hand column. X refers to the question in the design specific tool, see S5. *Abbreviations.* NHLBI: National Heart, Lung, and Blood Institute's Quality Assessment.

**Table 3. Interventions promoting occupational balance in adults.**

| Intervention | Target group | Goal | Features (duration, setting, format, leader) | Content |
|---|---|---|---|---|
| *Redesign your EVEryday Activities and Lifestyle with Occupational Therapy (REVEAL(OT))* [60] | Persons with chronic pain | To target meaningful occupations, regular physical activity, and a healthy diet. | Individual and group (max.6 participants) In-person led by trained OTs added to a standard multidisciplinary chronic pain treatment | (a) Brief didactic presentations (occupation for health and well-being, benefits of daily physical activity, eating habits, occupational balance and time management, productivity/ domestic and out-of-home activities, ergonomics at home and work, flow experience, hobbies, and leisure); (b) Group discussions; (c) Individual reflection prompts; (d) Building up personal experience. In addition, participants monitor their lifestyle-related health behavior by making diary notes and wearing an activity tracker which detects daily physical activity, energy expenditure and step counts. |
| *Leisure intervention* [69] | Persons recovering from substance use | To foster an understanding of preferred and accessible leisure activities, ultimately facilitating the process of leisure planning and engagement. | 2 months, twice a week, 12 sessions Group (9 participants) In-person Led by OT | Initial two and final sessions focus on education and planning, familiarizing participants with leisure concepts, identifying barriers and facilitators to leisure, understanding preferred and available leisure activities, and developing personalized leisure plans. Sessions 3–11 involve practical activities where the group engages in chosen leisure occupations together (including cinema visits, escape room experiences, walking, air hockey, museum visits, bowling, paintball, shooting, and karting). |
| *Balancing Everyday Life (BEL)* [37,53,54,66] | Persons with diagnosed mental illness and who want to develop a meaningful and balanced everyday life | To gain the ability for self-analysis in relation to everyday activities and to gain strategies for changing one's life. | 12 sessions and 2 booster sessions (1,5−2 hrs/session) in 16 weeks Group (5–8 participants) In-person Led by 1–2 therapists (at least one is OT) | Session's structure: Education, discussion, preparing for home assignment. Phase a) (sessions 1–3) "Intro and exploring occupation": exploring one's past and present occupational engagement, learning about occupational balance and imbalance, sources of meaning, purpose and motivation in life. Phase b) (sessions 4–12) "Better balance": working toward better balance, weekly topics include the art of rest and relaxation, mindfulness, nutrition, physical exercise, leisure activities, social life and relationships, productivity. Phase c) (2 booster sessions) "Summarizing and working independently": transition to working on one's own, reflection on progress made in order to prioritize what participants wanted to work on after the course |
| *The Tree Theme Method (TTM)* [39,55,56,68] | Persons with diagnosed depression and/or anxiety | To increase the individual's ability to cope with everyday life, and its intention is to develop strategies for becoming active, thereby enhancing patients' satisfaction with the mix of activities and routines that compose their everyday lives. | 5 sessions (1hr/session) during a period of 6–9 weeks Individual In-person Led by trained OT | The *TTM* starts each session with relaxation, and then the patient paints a tree representing a specific period of life: the present, childhood, adolescence, and adulthood. Based on the paintings, patients tell their occupational life story. At the end of each session, the patient and the OT have a reflective dialogue about necessary changes in the patient's everyday life. In the last session, the focus is on story making and on shaping plans. |
| *Time use intervention* [57] | Isolated persons due to coronavirus disease | - To maintain health and well-being by properly distributing time within the occupation area to maintain occupation balance and to plan a daily routine to engage in meaningful occupations. | 15 minutes/day, over 7 days Individual In-person Led by OT | Initial education on self-activity; time-use analysis (analyze time spent, list activities); occupation selection (based on K-LBI); activity assignment (place meaningful tasks in the meaningless time); practice and checkup (performing occupation individually and occupational therapy once a day for 15 minutes); creation of individual timetable |
| *Psychoeducational program* [61] | Persons with substance use disorder | - To highlight the importance of personal volition and the development of healthy, productive and meaningful lives and to support recovery | Once weekly (1 hr/session) Group In-person Led by OT | Psychoeducational sessions with information, group discussion, worksheets, goal setting, exercises, reflection and recovery planning. Topics: Stress and stress management, lifestyle balance, self-care, leisure and motivation. |

*(Continued)*

 

**Table 3.** (Continued)

| Intervention | Target group | Goal | Features (duration, setting, format, leader) | Content |
|---|---|---|---|---|
| *Ballroom dancing classes* [70] | Students | To reduce stress and promote occupational balance. | 16 sessions, twice a week (1 hr/session) Group (18 participants) In-person Led by OT | Each ballroom dancing class is structured into 5 moments: initial body stretching, teaching of movements, application of dynamics to favor learning, practice the rhythm worked, and closing the class. |
| *Occupation-based sleep program* [59] | Persons with diagnosed insomnia disorder | - To promote awareness of sleep hygiene factors and environment and to restructure participation in daytime activities with a focus on occupational balance. | Weekly sessions (2 hrs/session) over a period of 8 weeks Group (4–6 participants) and individual In-person Led by OT | 1)Sleep education: occupational level (review sleep pattern, identify factors affecting sleep, discuss occupational balance, goal setting), 2)Sleep education: personal level (integration in daily routine, psychological aspects, coping strategies, experience different calming activities, goal setting) 3)Sleep education: environmental level (key factors of sleep promoting environment, explore characteristics of occupation, sleep aids) 4)Integration of knowledge |
| *Web-based time use intervention* [58] | Students during the Covid-19 pandemic | To promote occupational balance of university students during the Covid-19 pandemic | 8 sessions (45 min/session), twice a week for a month Individual Web-based Led by OT | 1)General principles of time management (pandemic, occupational balance, health, prioritizing, time consumers, sleep/self-care(productivity/leisure activities); 2)Sleep (effects of sleep on other occupations, excessive/inadequate sleep); 3)Timetabling (effective time use strategies); 4)Self-care/leisure occupations (allocate time for these); 5)Productivity occupations (Prioritizing and schedule according to highest energy level during the day); 6)Establishing a new routine; 7)Time consumers (e.g., setting a quota for social media); 8)Maintenance of course leading (strategies for continuing) |
| *ReDesigning Daily Occupations (ReDO)* [67] | Women at risk of or on sick leave | To increase participants' understanding of the connection between their doing and their health. | Sessions twice a week (2,5 hrs/session) for 16 weeks *(ReDO-16)*/ 10 weeks *(ReDO-10)* Group (6–8 participants) In-person Led by OT | 1)Introduction; 2)Occupational history (personally meaningful occupations); 3&4)Occupational balance; 5&6)Patterns of daily occupations and time (exploration of how time is used, departing from a diary); 7&8)Hassles and uplifts in daily life (identifying and sharing in group); 9)Goal setting; 10)Occupational value (seminar, setting goals and prioritizing); 11)Evening seminar for friends, family, partners or employers (to introduce key principles of the pogram and process of change); 12&13)Goals and strategies; 14)Follow up |
| *Swedish Version of Let's Get Organized (LGO-S)* [66] | Persons with confirmed or suspected mental or neurodevelopmental disorder | To foster the development of effective time management habits and organizational skills. | 10 weekly sessions (1,5 hrs/session) Group (6–12 participants) In-person Led by OT | Structured training in the use of cognitive assistive techniques (e.g., use of a calendar) and building cognitive and emotional strategies using trial-and-error learning strategies for daily time management. Themes: managing time and the consequences of impaired daily time management on everyday activities, circadian rhythm, energy level during the day. Every session starts with a reflection on current mood. |
| *Action Over Inertia (AOI)* [34] | Persons with diagnosed schizoaffective disorder or schizophrenia | To improve occupational balance and engagement in meaningful activities to promote health and wellbeing. | One visit per week over a period of 12 weeks Individual In-person, home-based Led by trained OT | 1)Determining the need for change and securing investment in the change process; 2)Reflecting on current occupational balance and engagement patterns with rapid introduction of and support for meaningful activities; 3)Providing information about the relationship between serious mental illness and occupational balance and engagement; 4)Long-term goal planning and support; 5)Ongoing monitoring and refinement of plans. |

*Note.* Interventions are listed according to their last evaluation. K-LBI: Korean Version of the Life Balance Inventory; OT: Occupational therapist.

The main limitations of the included studies were small sample sizes and the absence of control groups, which limited the ability to robustly evaluate the effectiveness of the interventions. Other limitations included lack of follow-up assessments, reliance on unblinded evaluations, use of self-reported questionnaires, and potential participant bias – all of which may affect the reliability and the generalizability of the findings. The results of the critical appraisal were not used as exclusion criteria but were considered as potential explanations for divergent results and were incorporated to support the interpretation of the overall findings.

## Interventions addressing occupational balance

The 18 included studies assessed 12 different interventions: *ReDesign your EVEryday Activities and Lifestyle with Occupational Therapy (REVEAL(OT))* [60]*, leisure intervention* [69]*, Balancing Everyday Life (BEL)* [37,53,54,66]*, Tree Theme Method (TTM)* [39,55,56,68]*, time use intervention* [57]*, psychoeducational program* [61]*, ballroom dancing classes* [70]*, occupation-based sleep program* [59]*, web-based time use intervention* [58]*, ReDesigning Daily Occupation (ReDO-10)* [67]*, Let's Get Organized – Swedish version (LGO-S)* [65]*,* and *Action Over Inertia (AOI)* [34]. Details of the interventions are reported in Table 3.

While only one intervention was delivered web-based [58], all other interventions were performed in-person ($n = 11$). Most of the interventions were carried out in clinical or rehabilitation settings ($n = 9$). Remaining interventions were implemented in a university context ($n = 2$) or at home ($n = 1$) [34]. Six interventions used a group-based format [37,53,65–67,61,69,70], two interventions included both individual and group-based sessions [60,59] and four were conducted individually [34,40,55,56,68,58,57]. In the group-based interventions, the group size ranged from four participants in the *occupation-based sleep program* [59] to 18 participating in the *ballroom dancing classes* [70]. All interventions were led by at least one occupational therapist. However, three publications investigating the *BEL* intervention involved an additional professional, such as a nurse or a social worker, as a co-leader [37,54,66].

The duration of interventions ranged from one week [57] to 16 weeks [1,37,53,54], comprising five [39,55,56,68] to 14 sessions [37,53,54,66], and had a duration from 15 minutes [57] to 2,5 hours [67]. Eighty-four percent of the studies recruited persons with a specific diagnosis, mainly mental health disorders. These were individuals with anxiety/bipolar/depressive disorders in the *BEL* [37,53,54,66]*, TTM* [39,55,56,68]*, LGO-S* [65] and *AOI* [34] interventions, and persons with substance use disorder in the *psychoeducational program* [61] and *leisure intervention* [69]. One intervention each focused on persons with chronic pain [60], individuals with diagnosed insomnia disorder [59], isolated persons due to coronavirus disease [57] and women at risk for or on sick leave [67]. Two interventions, the *web-based time use intervention* [58] and *ballroom dancing* [70], were directed at students.

In terms of intervention content, most interventions ($n = 9$) included educational themes, providing theoretical input on topics such as occupation for health, well-being, benefits of activity, nutrition, occupational balance, time management, ergonomics, rest, relaxation, mindfulness, exercising, leisure activities, self-activity and sleep [37,53,54,66,60,61,69]. Seven interventions used self-reflection exercises to address changes in daily lives [34,39,55,56,65,60,61,67,68,69]. Identifying strengths and limitations in everyday life was part of the content of *TTM, ReDO* and *REVEAL(OT)* [39,55,56,68]. Group activities were employed in three interventions [37,53,54,66,61,69]. Home assignments to be completed in between sessions were employed in *BEL, TTM* and the *time use intervention*. Four interventions implemented individual goal setting [34,67,61,59]. As described in the analyzed publications, the transfer to the participants' everyday lives was addressed in 10 out of 12 interventions: building up personal experience [60], develop personal plans [69], transition to working on one's own [37,53,54,66], shaping plans [39,55,56,68] creation of individual timetables [57], planning [61], maintenance [58], strategies [65,67] and refinement of plans [34].

## The effectiveness of interventions targeting occupational balance

This section describes the key results regarding the effectiveness of the interventions, with more details on the study characteristics in Table 1 and the assessment of their quality in Table 2. Four publications utilized data from the same RCT

to explore the *Balancing Everyday Life intervention,* including 226 individuals in clinical and community-based settings at baseline [37,53,54,66]. The article published in 2017 described that *BEL* participants showed significant increases in activity engagement (p<0,001), activity level (p=0.036), general activity balance (p=0.042), reduction of symptom severity (p=0.046) and psychosocial functioning (p=0.018) compared to the control group [37]. Findings from Hultqvist et al. [66] indicated that having a close friend predicts clinically important improvement in occupational balance (p=0.023). Regarding recovery improvement, *BEL* was found to be equally beneficial and effective compared to standard occupational therapy in Argentzell et al.'s analysis in 2020 [53]. Eklund et al. [54] reported clinically significant improvement of occupational engagement (POES) in the *BEL* group at completion and at 16 weeks follow up (p=0.0004). However, one limitation that needs to be considered when interpreting the results is the significantly higher dropout rate in the experimental group compared to the control group, with participants in the experimental group dropping out mainly due to non-compliance with the intervention [37].

Four publications from analyses from one and the same study cohort included in this review explored the effectiveness of the *Tree Theme Method*, including 121 adults with depression and/or anxiety at baseline. The results of the original RCT conducted in 2018 indicated that both the experimental and control group improved on all outcomes measured from pre- to post-intervention. The improvements in occupational balance scores (COPM and OBQ) were statistically significant (p≤0.01) in both groups, indicating that the intervention was not significantly better than regular occupational therapy [55]. Furthermore, both the *TTM* and the control group showed long-term changes in their occupational balance, with the participants in both groups having a statistically significant improved score (p≤0.01) on the COPM and the OBQ after 3 and 12 months respectively [55]. The cohort study by Wagman et al. [68] analyzed part of the study population from the previous RCT based on their work situation and found that the occupational balance scores did not differ between the groups across all measurement points. Hakansson and colleagues [39] found that a high score on the "Balance between energy-giving and energy-taking occupations" item of the OBQ immediately after the intervention was a predictor of satisfaction with daily occupations 12 months later.

While *BEL* and *TTM* were analyzed multiple times, all other described interventions were each examined in only one of the included studies, as shown in Table 1. A significant improvement of occupational balance was achieved by the *leisure intervention* [69], *time-use intervention* [57], *psychoeducational program* [61], *ReDO-10* intervention [67], *LGO* intervention [65] and the *AOI intervention* [34]. The participants of the *web-based time use intervention* [46] significantly improved scores on all but one item of the OBQ. The students taking part in the *ballroom dancing intervention* [58] reported improved occupational balance compared to pre intervention. The results of the study on the *occupation-based sleep program* [59] showed that the changes in occupational balance of the experimental group were significantly higher than those of the control group. In some cases, control group participants who received a shorter version of the intervention [58] or standard care [34], also improved occupational balance scores.

Due to the variability of data and measurement instruments across studies, no meta-analysis was conducted and therefore no correlations were calculated or reported in this systematic review. Quality of life was the most common additional outcome measured in relation to occupational balance (*n*=8). Other secondary outcomes included functioning (e.g., mastery, performance of activities, motor and process skills) and health-related measures based on diagnoses and symptoms (e.g., pain severity, insomnia index, symptom checklists, health status and/or recovery). All secondary outcome measures and their measurement instruments are listed in Table 1.

**Sustainability of results.** As shown in Table 1, ten studies indicated a long-term effect of six interventions (*leisure intervention, BEL, TTM, occupation-based sleep program, ReDO, LGO*), with follow-ups ranging from two [69] to 12 months [39,56,68]. Ongoing improvements could be shown in occupational balance, personal recovery, engagement in activities, psychological symptoms, health-related aspects, organization and planning skills [39,55,56,67,68,59]. Common characteristics of these interventions include a group-based and in-person format, a duration of six [39,55,56,68] to 16 weeks [37,53,54,66], and delivery by an occupational therapist.

## Discussion

This research aimed at supporting health professionals, researchers, and policymakers to understand and critically reflect the current body of peer-reviewed evidence on interventions targeting occupational balance. It is the first systematic literature research synthesizing and assessing interventions that address occupational balance in diverse adult populations.

**Interventions addressing occupational balance.** The interventions included in this review shared several common characteristics: they were largely delivered in-person, tended to target people with mental health problems, and employed a variety of methods to promote occupational balance and support participants in transferring these changes to their daily lives.

The mode of delivery of most interventions was group-based and previous research has shown that therapeutic group settings are indeed beneficial for participants [71]. Given the importance of critically examining one's current lifestyle in achieving occupational balance [72], group-based interventions and group introspection may be particularly valuable for enhancing occupational balance.

This review identified two interventions that were carried out during inpatient treatment. While the goal of improving occupational balance of patients, who are hospitalized for an indefinite period of time, seems reasonable, it is so far unclear whether occupational balance can be sustainably enhanced in inpatient contexts shaped by hospital or clinic routines far off a person's actual everyday life. It may be assumed that interventions are most effective when carried out in home environments, since a positively experienced occupational balance may, among other factors, be achieved through changes in everyday routines [73]. Only one virtual intervention addressing occupational balance was identified in this review [58], which is surprising given the recent acceleration of digital intervention trends [74]. Considering the need for a strong link between interventions addressing occupational balance and individuals' daily lives, digital interventions may be an effective delivery mode. This is because users of these digital interventions may be able to more easily integrate changes into their daily routines and activities, regardless of their physical environment, making it a promising approach [75]. Future research should investigate whether digital solutions truly deliver this benefit.

While the content of the 12 included interventions varied to some degree, common themes were addressed, including education and self-reflection about occupational balance and individual use of time, goal setting related to occupational balance, and implementation of habits into daily life. Further aspects related to the concept of occupational balance that were covered in the interventions included eating habits, ergonomics, self-care, time consuming activities, use of creative techniques, dancing, body stretching, calming activities, sleep education, and the use of a diary. This aligns with earlier evidence and conceptualizations of occupational balance, where factors such as activity balance, balance in body and mind, mindfulness, self-awareness, relaxation, balance in relation to others, organization of time, and time balance were identified as relevant for adult's occupational balance [5]. However, considering that individually meaningful occupations play an important role in the concept of occupational balance [32], it is worth questioning whether interventions such as creative work, dancing or calming activities can contribute to occupational balance. In a study, Yazdani et al. [76] explored how the concept of occupational balance is perceived and practiced by occupational therapy practitioners and identified a distinction between meaningful and purposeful occupations. While the former hold personal significance for the individual, the latter refer to occupations that are beneficial to engage in but may not necessarily be of personal value to the individual. In line with earlier research, the results of this review stress the importance of not only implementing meaningful, but also purposeful occupations to achieve therapy goals [76]. In particular, group settings may benefit from this approach, acknowledging that although large groups cannot provide fully individualized strategies, they can still gain from occupations that are generally perceived as helpful in supporting occupational balance.

Additionally, sleep is critical in relation to occupational balance [77], as maintaining a balance between rest/sleep and daytime activities is essential to promote function and well-being [72,78,79]. This aligns with research highlighting the health benefits of adequate sleep [80,81]. The limited focus on sleep in current interventions addressing occupational balance may be attributed to insufficient evidence on the effect of occupational therapy on sleep [82] and the ambiguity of whether sleep can be defined as an occupation [77].

Most of the studies collected data on social relations, but few explicitly analyzed how these influenced intervention outcomes. Only the articles describing *ReDO* and *BEL* reported the consideration of the social dimension, which was not reflected in the studies' outcome measures. In light of previous evidence emphasizing the importance of social relationships for balance in daily life [66], this appears noteworthy. It could be interpreted that this represents a potential omission in the included studies, or that a possible reason for not integrating the social dimension could be related to challenges in assessing it. Nevertheless, since social relationships are an essential part of human life [13], it seems important to consider social dimensions more strongly in both the design and evaluation phases in future research targeting occupational balance.

In addition, the diverse conceptualizations of occupational balance may explain the heterogeneity of the interventions. A need for a stringent definition of occupational balance to better distinguish it from other concepts has been identified in former research [6]. Occupational balance encompasses more than the dichotomous rationale as found in the work-life balance concept, describing the management of paid work and the rest of life [83]. The various components of occupational balance should accordingly be clearly reflected when designing interventions targeting occupational balance.

**Effectiveness of interventions targeting occupational balance.** The results indicate that some interventions significantly increased occupational balance scores and improved activity levels, symptom severity, and psychosocial functioning, defined as the individuals' psychological, social, and occupational performance [84]. However, due to wide variation in measurement instruments, study designs, and study quality, results should be interpreted with caution. For example, one study showed significant improvements in occupational balance from pre- to post-intervention but included a small sample size [69]. As another example, two studies that reported beneficial effects for the *AOI intervention* and the *ballroom dancing intervention*, were rated as low quality due to small sample sizes, significant dropout rates and the use of a non-validated outcome measure [34,70]. Deciding which measure to use to assess occupational balance seems to be difficult, as evidenced by the wide range of outcome measures across all 18 articles. Scholars have previously argued that measuring occupational balance is particularly challenging due to the complexity of the concept and its variety of definitions [6,85], compromising the comparability of studies.

Apart from statistically significant effects of the explored interventions, several studies reported clinically significant effects in terms of improvements in occupational balance, indicating their potential to enhance individuals' overall health [34,56,65,57,86]. Clinically important long-term improvements in satisfaction with daily occupations, psychological symptoms of anxiety and depression, and health-related aspects were shown in *TTM* [55,56], while *LGO* demonstrated clinical utility in improving occupational balance and engagement for people with serious mental disorders. Initial positive data on the efficacy and clinical utility of the *AOI* intervention were obtained in the included pilot study [34]. As clinical significance refers to the extent to which an intervention makes a tangible and meaningful difference in the daily lives of patients or those with whom they interact [86,87], it is inarguable that not only the quantifiable changes in occupational balance measures are worth mentioning, but also subsequent benefits in other aspects of life. For example, improved occupational balance also reduced drug craving and enhanced leisure participation in individuals with substance use disorders [69]. In isolated patients, improved occupational balance led to better scores in mental health and quality of life [57]. Additionally, improved outcomes affected work ability in women with depression [67].

**Gender and cultural dimensions of decreased occupational balance.** Experiencing an occupational imbalance can be seen as an overarching theme in the selection of included populations. Study participants in this review cannot be perceived as representative of the general adult population, as there are some noticeable trends. First, there is a clear predominance of female participants. Second, the average age of the participants(40–57 years) was within the working age range, lacking data on older populations. Third, it is noteworthy that 12 out of 18 studies focused on people with mental health problems. These findings go in line with a scoping review exploring current research on occupational balance [32], in which the authors point out that the predominance of female participants may also be related to the respective diagnoses defined as inclusion criteria. Especially the gender aspect requires further examination as it remains

unclear if women experience or report occupational imbalance more often, or if chosen methods lead to selection or participation bias. Potential explanations for women's imbalance could imply women's tendency to have double workload and a more complex pattern of occupation then men [67]. For example, it is usually women who spend twice as much time as men on care work enabling health systems functionality when it comes to informal care of persons in need for assistance in their homes [88] or social systems functionality when thinking of childcare obligations [89]. This load can lead to serious health risks, such as anxiety, depression, loneliness and occupational imbalance [21,90–92].

It is important to note that all included studies were conducted in high-income countries only, with a majority being conducted in so called "western societies". Considering the assumption that decreased occupational balance is a concern mainly the more privileged populations can think about [93,94], it is crucial to further examine whether cultural differences shape the conceptualization, measurement, and implementation of occupational balance interventions.

**Implications for research and clinical practice.** The rapidly accelerating and changing everyday life due to digitalization, the experience of lock-down measures during the COVID-19 pandemic, as well as neoliberal political trends towards individual responsibility and privatization versus state involvement have put the concept of occupational balance under the spotlight. Hence, public health experts, policymakers and health scientists become increasingly aware of the negative effects of occupational imbalance, jeopardizing health and wellbeing [95–98]. Scholars agree that interventions addressing occupational balance may be promising from a public health perspective [13,34]. While some interventions followed a community-based approach, most of them have been explored in clinical in- or outpatient healthcare settings or were delivered by institutions outside the healthcare sector, e.g., universities. Future research should supplement existing interventions with more community-based, low-threshold services, or digital solutions. Expanding the scope to include more diverse and accessible formats will help increase their public health impact.

Furthermore, existing interventions primarily address occupational balance from an individual perspective. Since research has shown that state-driven policy measures can improve work-life balance [99], it can be assumed that they may also facilitate societal-level changes and help mitigate decreased occupational balance. Adjusted legislation for parents, individuals with chronic illnesses and informal caregivers, as well as barrier-free solutions for people with disabilities would enable individuals to experience greater balance and meaning in their everyday activities. To promote occupational balance among diverse population groups, the implementation of flexible working solutions, improved childcare services, tailored support for individuals on long-term sick leave, and customized support for informal carers appears promising.

**Strengths and limitations.** The strengths of this review include adherence to the PRISMA guidelines, a registered protocol, consideration of the RE-AIM framework, and diligent appraisal of study quality. To our knowledge, this is the first review on interventions that address occupational balance in adults regardless of health status or setting. Some limitations of the review must be considered. Due to the chosen methodology, interventions studied using qualitative or mixed-method designs were not included in this review. Acknowledging the subjectiveness of the occupational balance concept as well as the fact that it may differ across cultures, age groups and populations is crucial. Future projects aiming to design occupational balance interventions may follow participatory research approaches and truly involve the population of interest throughout the design process. In fact, additional interventions promoting occupational balance were detected, which still need to be examined for their effectiveness with quantitative methods. Among the interventions discovered were *Project Bien Estar* [100], *self-management occupational therapy program (SMOoTh)* [101], *educational workshop on time use* [102], *inpatient energy management education (IEME)* [103], *mindful based program* [104], *home modification intervention* [105], *psychological rehabilitation program* [106], *therapeutic gardening* [33] and *Daily Life Coping [*107*]*. Given that these interventions demonstrate promising approaches and existing research may already incorporate the aforementioned participatory methods, further investigation is warranted. As the researchers expected a great heterogeneity of measurements and research designs, a meta-analysis procedure was considered as inappropriate, and a narrative synthesis approach was used. A more homogeneous use of measurement instruments would enable more comprehensive analyses.

## Conclusions

Our systematic literature review demonstrated a wide range of interventions developed to enhance individuals' occupational balance. The heterogeneity and diversity of reviewed interventions have been reflected in their scope and purpose, conceptualization of occupational balance, study designs, settings and target groups.

Several interventions have proven effective in improving occupational balance and secondary outcomes, potentially enhancing the health, well-being, and life satisfaction of adults. Occupational balance interventions can complement health approaches in a variety of settings, such as clinical environments, workplaces, schools, or community-based institutions. Implementing these interventions would enable occupational therapists to broaden their scope of action, complementing other professions and public health approaches.

There is still a need for more detailed evaluations of interventions promoting occupational balance. Future studies should employ controlled experimental designs to assess interventions in diverse populations and larger samples, target gender- or age-related differences, and provide high-quality evidence for effectiveness across various delivery modes and settings.

## Supporting information

**S1 Table. Prisma checklist.**
(DOCX)

**S2 Table. Amendments to protocol.**
(DOCX)

**S3 Table. Search string.**
(DOCX)

**S4 Table. All studies identified in the literature search.**
(DOCX)

**S5 Table. NHLBI quality assessment tools.**
(DOCX)

**S6 Table. Data extraction.**
(XLSX)

## Acknowledgments

Thanks to Michael Schön (Duervation), who provided feedback to the manuscript's first draft.

## Author contributions

**Conceptualization:** Stefanie Lentner, Evelyn Haberl, Larisa Baciu, Mona Dür, Cornelia Lischka, Mandana Fallahpour, Susanne Guidetti, Hanna Köttl.

**Formal analysis:** Stefanie Lentner, Evelyn Haberl, Hanna Köttl.

**Funding acquisition:** Mona Dür, Mandana Fallahpour, Susanne Guidetti, Hanna Köttl.

**Investigation:** Stefanie Lentner, Evelyn Haberl, Larisa Baciu, Hanna Köttl.

**Supervision:** Mona Dür, Hanna Köttl.

**Writing – original draft:** Stefanie Lentner, Hanna Köttl.

**Writing – review & editing:** Stefanie Lentner, Evelyn Haberl, Larisa Baciu, Mona Dür, Cornelia Lischka, Mandana Fallahpour, Susanne Guidetti, Hanna Köttl.

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
