## [Decision Letter · Decision Letter 0]

15 Dec 2024

PONE-D-24-45446Interventions promoting Occupational Balance in Adults: A Systematic Literature ReviewPLOS ONE

Dear Dr. Lentner,

Thank you for submitting your manuscript to PLOS ONE. After careful consideration, we feel that it has merit but does not fully meet PLOS ONE’s publication criteria as it currently stands. Therefore, we invite you to submit a revised version of the manuscript that addresses the points raised during the review process.

While this manuscript offers a promising review of interventions promoting occupational balance in adults, it necessitates significant revisions to meet the standards of publication. Methodological clarity, articulation of clinical significance, and presentation of results require particular attention. Please refine your search strategy, elaborate on the real-world impact of interventions, and enhance the clarity and organization of your findings to strengthen the manuscript's potential for impact and contribution to the field.

We look forward to receiving your revised manuscript.

Kind regards,

Denis Alves Coelho, PhD

Academic Editor

PLOS ONE

Journal Requirements: When submitting your revision, we need you to address these additional requirements. 1. Please ensure that your manuscript meets PLOS ONE's style requirements, including those for file naming. The PLOS ONE style templates can be found at https://journals.plos.org/plosone/s/file?id=wjVg/PLOSOne_formatting_sample_main_body.pdf and https://journals.plos.org/plosone/s/file?id=ba62/PLOSOne_formatting_sample_title_authors_affiliations.pdf 2. As required by our policy on Data Availability, please ensure your manuscript or supplementary information includes the following:  A numbered table of all studies identified in the literature search, including those that were excluded from the analyses.   For every excluded study, the table should list the reason(s) for exclusion.   If any of the included studies are unpublished, include a link (URL) to the primary source or detailed information about how the content can be accessed.  A table of all data extracted from the primary research sources for the systematic review and/or meta-analysis. The table must include the following information for each study:  Name of data extractors and date of data extraction  Confirmation that the study was eligible to be included in the review.   All data extracted from each study for the reported systematic review and/or meta-analysis that would be needed to replicate your analyses.  If data or supporting information were obtained from another source (e.g. correspondence with the author of the original research article), please provide the source of data and dates on which the data/information were obtained by your research group.  If applicable for your analysis, a table showing the completed risk of bias and quality/certainty assessments for each study or outcome.  Please ensure this is provided for each domain or parameter assessed. For example, if you used the Cochrane risk-of-bias tool for randomized trials, provide answers to each of the signalling questions for each study. If you used GRADE to assess certainty of evidence, provide judgements about each of the quality of evidence factor. This should be provided for each outcome.   An explanation of how missing data were handled.  This information can be included in the main text, supplementary information, or relevant data repository. Please note that providing these underlying data is a requirement for publication in this journal, and if these data are not provided your manuscript might be rejected. 3. Thank you for stating the following financial disclosure: "This project is funded by the Gesellschaft für Forschungsförderung Niederösterreich m.b.H. (GFF) as part of the RTI-Strategy 2027 (Grant: FTI21-P-005). The funder's website: https://www.gff-noe.at/. GFF had no influence on the research or publication process.This review is part of the CROB project (Collaborative Research on Occupational Balance), which is a research collaboration between the IMC University of Applied Sciences Krems (Austria), Duervation (Austria) and Karolinska Institutet (Sweden)." Please state what role the funders took in the study.  If the funders had no role, please state: ""The funders had no role in study design, data collection and analysis, decision to publish, or preparation of the manuscript."" If this statement is not correct you must amend it as needed. Please include this amended Role of Funder statement in your cover letter; we will change the online submission form on your behalf. 4. Please include captions for your Supporting Information files at the end of your manuscript, and update any in-text citations to match accordingly. Please see our Supporting Information guidelines for more information: http://journals.plos.org/plosone/s/supporting-information. 

**Additional Editor Comments:**

Your work addresses an important topic, and both reviewers found it interesting and relevant. However, they have raised several concerns that need to be addressed before your manuscript can be considered for publication. Reviewer 1 has requested a minor revision, while Reviewer 2 suggests a major revision. Both reviewers have provided detailed comments and suggestions for improvement, which you should carefully consider. Both reviewers emphasized the need for clearer descriptions of the interventions and their impact. Please elaborate on the clinical significance of the findings and provide more context around the outcomes. Reviewer 1 raised questions about the search strategy, selection criteria, and data extraction methods. Please clarify these aspects and ensure they are robust and transparent. Both reviewers suggested improvements in the presentation of the results, including the tables and figures. They also pointed out the lack of information on dropout rates and strength of correlations. Moreover, reviewer 2 noted some issues with the language and writing style, particularly in the abstract. Please revise the manuscript to ensure clarity and accuracy.

Reviewers' comments:

Reviewer's Responses to Questions

**Comments to the Author**

1. Is the manuscript technically sound, and do the data support the conclusions?

Reviewer #1: Yes

Reviewer #2: Partly

2. Has the statistical analysis been performed appropriately and rigorously? 

Reviewer #1: Yes

Reviewer #2: N/A

3. Have the authors made all data underlying the findings in their manuscript fully available?

Reviewer #1: Yes

Reviewer #2: Yes

4. Is the manuscript presented in an intelligible fashion and written in standard English?

Reviewer #1: Yes

Reviewer #2: No

5. Review Comments to the Author

Reviewer #1: Summary:

This manuscript was a systematic review of the literature that looked at occupational balance. The study extracted details from 18 studies, many of which were from Sweden, and the rest from other countries. Overall, this was an interesting manuscript on an important topic. The detail provided in the tables was easy to read. My question that remained at the end of reading this was one of clinical significance. Statistical significance was addressed throughout, but a value can gain statistical significance without having any meaning in the real world. Did any of the included studies report any clinically significant changes to their population, following the administration of their intervention? I believe the clinical significance of achieving OB may be necessary to be built in throughout.

The points I have listed below relate to specific areas of the manuscript.

Questions about specific sections:

Abstract. This does not seem to be written with the same level of care and language as the rest of the manuscript. Some phrasing is difficult to understand, and the definition of OB is challenging. The results are quite non-specific, and the conclusion is a bit vague on details as to whether this is important or not.

Manuscript.

Line 80 ref (23) Is this the right one?

L 81 past tense – potentially affect

L105 have (not has)

L106 Don’t start sentence with Especially. The sentence is unclear in its meaning.

L111-115: Context getting a bit mixed

L 116-118 need rewording. Quite confusing.

L121-123: research questions don’t align with clinical context mentioned earlier.

Selection criteria: Not sure why point two was added, if it didn’t matter? Is this maybe a data extraction item?

L143: Why were the databases searched over a 2 month period, instead of all on the same day, to make sure you got consistent reports from the databases?

L149 – why searched for in German if only English language was included (L138)?

L 155 – what was deemed a completely irrelevant record and who made that decision?

L168-173: Can this info please be added to the supplementary material with the questions.

L160 & 175: What was the value of using the kappa statistic? Was further discussion entailed if too low between raters?

Study flow chart: No mention of the number of duplicates found and the number of irrelevant records.

Selection criteria did not say that protocols were not accepted – is this because it is assumed that no values were available for the outcomes measures?

L199 – the reasons given for exclusion don’t seem to match what is presented in the diagram?

L221: Mean age??

L221: years since diagnosis: this isn’t in the extraction criteria? And earlier said inclusion could be with and without a condition?

L225: This makes sense of the random clinical mention in introduction now. Can this important information, including the definitions be moved to the introduction and/or methods. This appears to come out of nowhere.

Table 1: How are the studies organised in the table? They don’t seem to be alphabetical, and pre-posts are before the RCTs? Found it at the end of the table -> do you mean the intervention is alphabetical? How does this relate to the narrative synthesis approach taken described in the methods?

T1: Time/length of intervention not always reported or reported as missing.

T1: Farhadian: is it possible to conduct an analysis on 9 people? Does sample size calculation stand up?

T1: How many of the Swedish studies are analyses from the same cohort, performed over different time periods or relating to different outcomes? If any, can they be synthesised so as not to give the impression they are different populations.

Can Table 2 be colour coded or another visual method use to show the differences between the ROB categories? Can headings also be added to show which studies are the same design, instead of a,b and c. Also, consider using N/A instead of empty cells, to show that this was not applicable, rather than forgotten.

Table 3: does the a with the accent indicate a mean value? Or approximately?

Heading on L286 is followed by a paragraph about how the outcomes were measured. The followed by whether results were positive or not, but not the extent of the change. Difficult to assess whether change was clinically significant.

In results, strength of correlations are not documented in paragraph starting 319.

Where are the drop-out rates listed?

In discussion, the extent of change, and whether it is of any clinical significance, is not discussed?

Paragraph starting L346: can this be reworded to address the contents of the paragraph, instead of a reference back to the results.

I am missing the bit about where an improvement in OB for these populations meant something to their overall health? What was the target score for these populations?

Reviewer #2: Please see attached file.

My conclusion is that the paper addresses an interesting topic of relevance for researchers and staff involved in activity-based rehabilitation. There are several issues that need to be addressed, however, and although the manuscript has potential it is not suitable for publication in its present form. I believe the authors have the ability to rework the manuscript according to the comments above encourage them to resubmit a revised version.

6. PLOS authors have the option to publish the peer review history of their article (what does this mean? ). If published, this will include your full peer review and any attached files.

**Do you want your identity to be public for this peer review?** For information about this choice, including consent withdrawal, please see our Privacy Policy .

Reviewer #1: No

Reviewer #2: No

---

## [Author Response · Author response to Decision Letter 1]

20 Feb 2025

Dear Reviewers and dear Editor,

Thank you for your valuable feedback on our manuscript. We appreciate the time and effort you have taken to review our work. We have thoroughly implemented all your comments and believe that your input has significantly improved the manuscript. Below, we have addressed each of your comments in detail.

Reviewer 1:

1. Comment: Abstract. This does not seem to be written with the same level of care and language as the rest of the manuscript. Some phrasing is difficult to understand, and the definition of OB is challenging. The results are quite non-specific, and the conclusion is a bit vague on details as to whether this is important or not.

Response: We are incredibly grateful for this important comment. Based on this feedback, we have spent further time on improving the abstract and very much hope that this latest version more strongly emphasizes the relevance and clinical significance of our systematic literature review. (lines 2ff, page 1&2)

2. Comment: Line 80 ref (23) Is this the right one?

Response: Thank you for pointing this out. We have added the correct reference (line 78, page 4).

3. Comment: L 81 past tense – potentially affect

Response: Thank you for noticing the tense mistake, we have changed it accordingly (line 79, page 4).

4. Comment: L105 have (not has)

Response: Thank you for raising this grammar mistake. (line 100, page 5)

5. Comment: L106 Don’t start sentence with Especially. The sentence is unclear in its meaning.

Response: Thank you for making us aware of the lack of clarity. We fully agree with reviewer 1 that this sentence seemed out of context and have hence deleted it.

6. Comment: L111-115: Context getting a bit mixed

Response: Thank you for raising this problem. We have accordingly changed the whole paragraph and hope that it is more targeted now. (lines 103ff, page 5)

7. Comment: L 116-118 need rewording. Quite confusing.

Response: Thank you for making us aware of this incongruency. We have now reformulated the gap of knowledge which has led us to delete this sentence.

8. Comment: L121-123: research questions don’t align with clinical context mentioned earlier.

Response: Thank you for this comment and for making us aware of this. We have further elaborated the contexts and added “in diverse contexts” in line 111, page 5. We hope that there is congruence now.

9. Comment: Selection criteria: Not sure why point two was added, if it didn’t matter? Is this maybe a data extraction item?

Response: Thank you for bringing this up. Our intention was to emphasize that both people with and without a diagnosis were included in this review. We have now changed the formulation to make it clearer that we “involved persons with and without any diagnoses” (lines 122&123, page 6).

10. Comment: L143: Why were the databases searched over a 2-month period, instead of all on the same day, to make sure you got consistent reports from the databases?

Response: Thank you for making us aware of this potential confusion. We mentioned the time frame for the entire search process, including the development of the search string, pilot searches and hand searches. The final search was carried out on April 8th, 2024. We have added this date now to the manuscript (line 134, page 6).

11. Comment: L149 – why searched for in German if only English language was included (L138)?

Response: Thanks for pointing this out. We forgot to mention this inclusion criterion here because we noticed that there were no German hits. We have now corrected this in the manuscript, added German to the selection criteria in lines 127&128, and stated in the “Study characteristics” section (line 211, page 9) that “The language of publication for all studies in the review was English”.

12. Comment: L 155 – what was deemed a completely irrelevant record and who made that decision?

Response: We used this to capture records that did not comply with the inclusion criteria. To enhance clarity, we removed “completely irrelevant records” (line 146, page 7).

13. Comment: L168-173: Can this info please be added to the supplementary material with the questions.

Response: Thank you for your suggestion. We have removed the information about the NHLBI quality assessment tools in this section and instead added it in the Supplementary Material as recommended (S5 table).

14. Comment: L160 & 175: What was the value of using the kappa statistic? Was further discussion entailed if too low between raters?

Response: Thank you for this input. We have better described the reasoning behind this in the method sections, lines 149f, page 7: “Cohen’s kappa coefficient was calculated for the screening process to evaluate the inter-rater agreement (44) and to initiate further discussion if the coefficient was too low.“ and added it accordingly in line 205, page 9: “further discussion about inclusion was not necessary.”

15. Comment: Study flow chart: No mention of the number of duplicates found and the number of irrelevant records.

Response: Thank you for pointing this out. We adjusted the flow chart (fig 1), mentioning the number of duplicates. To avoid misunderstanding we are no longer using the expression “irrelevant records”.

16. Comment: Selection criteria did not say that protocols were not accepted – is this because it is assumed that no values were available for the outcomes measures?

Response: Thank you for highlighting this. Our aim was to identify existing interventions that have already been tested, with findings published in peer-reviewed journals; therefore, we excluded study protocols. We have added examples to the exclusion criteria, to be more explicit on this point: “a) were not published in a German or English-language, peer-reviewed journal (e.g., study protocols, poster presentations)” (lines 128ff, page 6)

17. Comment: L199 – the reasons given for exclusion don’t seem to match what is presented in the diagram?

Response: Thank you for pointing this out. To be more explicit about the reasons for exclusion at each stage of the selection process, we have added “Records excluded based on the inclusion and exclusion criteria” in Fig 1 in the screening phase. We also added information about the reasons for exclusion at the eligibility stage, consistent with those stated in Fig 1, in the main text at lines 199ff, page 9: “Studies were excluded at this step because they were study protocols (n=5), trial registrations (n=7), incorrect study design (n=7) or non-peer-reviewed articles (n=4), or because occupational balance was not addressed (n=10).”

18. Comment: L221: Mean age??

Response: Thanks for pointing out this grammar mistake, we have changed “mean ages” to “mean age”, line 217, page 9.

19. Comment: L221: years since diagnosis: this isn’t in the extraction criteria? And earlier said inclusion could be with and without a condition?

Response: Thank you for making us aware of this inconsistency, since we included persons with and without diagnosis we added “if applicable” to make this clearer, also adding “if applicable: years since diagnosis” in line 166, page 7 under “data extraction and data analysis”.

20. Comment: L225: This makes sense of the random clinical mention in introduction now. Can this important information, including the definitions be moved to the introduction and/or methods. This appears to come out of nowhere.

Response: Thank you for this comment. We have now mentioned the different contexts of interventions earlier in the background section and further explained the definitions of contexts in the method section, see lines 169ff, page 8: “For this review, study settings were categorized into clinical, community-based and academic setting. The term clinical was used to refer to medical work relating to the examination and treatment of persons based on their health status, also involving rehabilitation, as the process of returning to a way of life after being ill (46). The notion community-based was defined as a setting that takes place locally, where an individual works, plays, and performs other daily activities (47). Academic settings were related to schools, colleges, universities or connected with studying (46).”

21. Comment: Table 1: How are the studies organised in the table? They don’t seem to be alphabetical, and pre-posts are before the RCTs? Found it at the end of the table -> do you mean the intervention is alphabetical? How does this relate to the narrative synthesis approach taken described in the methods?

Response: Thank you for raising this question. We have added the heading “Intervention” to the title of the first column and included the titles of interventions accordingly. Also, we have organized the publications by study design (starting with RCTs, followed by pre-post studies, and finally cohort studies) and then alphabetically. To enhance clarity, we have also revised the explanation in the notes: “Note. Studies are listed by study design and then alphabetically”, see pages 11-14.

22. Comment: T1: Time/length of intervention not always reported or reported as missing.

Response: Thank you for making us aware that the reader might search for this information in table 1. Since the focus of this table is on the characteristics of the study, we kept the information on length of intervention in table 3 “Interventions promoting occupational balance in adults”, page 17&18. To clarify for the reader, we added “Additional information on the interventions reported is summarized in table 3.” in lines 223f, page 10 in the body text and “Additional information on the interventions is reported in table 3” in the note for table 1, page 14.

23. Comment: T1: Farhadian: is it possible to conduct an analysis on 9 people? Does sample size calculation stand up?

Response: Thank you for bringing up this legitimate issue. To answer research question 1, we did not exclude studies based on the results of the critical appraisal, as described in lines 236ff, page 15. However, when answering research question 2, we do indeed discuss the problem related to such a small sample size. Based on your feedback, we have put stronger emphasizes on this, beginning from line 429, page 25f “…because the measurement instruments, study designs, and quality of the studies vary widely, some of the results must be viewed with caution. For instance, one study (66) showed significant improvements in occupational balance from pre- to post-intervention, but due to the small sample size of only nine participants the results should be treated with care.”

24. Comment: T1: How many of the Swedish studies are analyses from the same cohort, performed over different time periods or relating to different outcomes? If any, can they be synthesised so as not to give the impression they are different populations.

Response: Thank you for pointing this out. We tried to clarify this misunderstanding by pointing out that they are analyses from the same cohort both in table 1, page 11: “analyses from the same study cohort as in (37)” as well as in the body text, line 288, page 20.

25. Comment: Can Table 2 be colour coded or another visual method use to show the differences between the ROB categories? Can headings also be added to show which studies are the same design, instead of a,b and c. Also, consider using N/A instead of empty cells, to show that this was not applicable, rather than forgotten.

Response: Thank you for this suggestion, we have implemented a colour code to show different ROB categories and inserted headings for the study designs. Also, we inserted “n/a” instead of leaving empty cells to avoid misunderstandings, table 2 on page 15&16.

26. Comment: Table 3: does the a with the accent indicate a mean value? Or approximately?

Response: Thank you for raising this issue. We have now used a slash to indicate the duration per session to enhance readability and avoid misunderstandings, e.g. “1hr/session”, see table 3 on page17&18.

27. Comment: Heading on L286 is followed by a paragraph about how the outcomes were measured. The followed by whether results were positive or not, but not the extent of the change. Difficult to assess whether change was clinically significant.

Response: Thank you for this comment. Based on your feedback, we have changed the order and content of this section, starting from line 284 on page20.

28. Comment: In results, strength of correlations are not documented in paragraph starting 319.

Response: We added an explanation that strength of correlations were not calculated in this review (as few studies have actually calculated these calculations.). The previously discussed topic of possible correlations was changed into a more cautiously expression: “Due to the variability of data and measurement tools across studies, it was decided not to conduct a meta-analysis and therefore no correlations were calculated or reported in this systematic review.”, (line 330ff, page 22).

29. Comment: Where are the drop-out rates listed?

Response: Thank you for raising this issue. We have added all drop-out rates in table 1, pages 11ff.

30. Comment: In discussion, the extent of change, and whether it is of any clinical significance, is not discussed?

Response: Thank you for this valuable information. We have added this accordingly and now discuss the clinical significance in more detail in the section starting on line 449ff, page 26.

31. Comment: Paragraph starting L346: can this be reworded to address the contents of the paragraph, instead of a reference back to the results.

Response: Thank you for your comment. We edited this section to avoid referring back to the results (lines 356ff, page 23): “The interventions included in this review shared several common characteristics: they were largely delivered in person, tended to target persons with mental health problems, and employed a variety of methods to promote occupational balance and support participants in transferring these changes to their daily lives.”.

32. Comment: I am missing the bit about where an improvement in OB for these populations meant something to their overall health? What was the target score for these populations?

Response: Thank you for pointing this out. We added more content on this in the paragraph starting on line 449, page 26: “As clinical significance refers to the extent to which an intervention makes a tangible and meaningful difference in the daily lives of patients or those with whom they interact (84, 85), it is arguable that not only the quantifiable changes in occupational balance measures are worth mentioning, but also subsequent benefits in other aspects of life. For example, improved occupational balance also reduced drug craving and enhanced leisure participation in individuals with substance use disorders (66). In isolated patients, improved occupational balance led to better scores in mental health and quality of life (67). Additionally, improved outcomes affected work ability in women with depression (61).”.

Reviewer 2:

1. Comment: Neither the abstract nor the introduction presents the rationale for the study. Why is the review needed? Actually, this is best expression in the discussion, page 22, line 341 and the following, but it needs to be stated in the intro – and abstract – as well.

Response: Thank you for raising this important issue. We have now more strongly emphasized the rationale for this work both in the background and the abstract, see abstract and line 98ff, page 5.

2. Comment: Page 3, lines 54-58. This sentence is unclear and needs to be reformulated.

Response: Thank you for pointing this out. We have reformulated the sentence accordingly: “Various factors can contribute to an imbalance in daily occupations. These include lack of time to complete desired or necessary tasks, limited opportunities to manage how time is allocated across activities, inflexibility, a mismatch between desired and required activities, and having either too much or too little to do.”, lines 51ff, page3.

3. Comment: Under heading ‘Gap of knowledge…’, page 5, lines 111 and forward should include the rationale, but instead there is a reasoning that does not seem totally logical. It is true that occupational balance is seen as a dynamic phenomenon, but is it therefore a good idea to “examine occupational balance

---

## [Decision Letter · Decision Letter 1]

25 Mar 2025

PONE-D-24-45446R1Interventions promoting Occupational Balance in Adults: A Systematic Literature ReviewPLOS ONE

Dear Dr. Lentner,

Thank you for submitting your manuscript to PLOS ONE. After careful consideration, we feel that it has merit but does not fully meet PLOS ONE’s publication criteria as it currently stands. Therefore, we invite you to submit a revised version of the manuscript that addresses the points raised during the review process.

We look forward to receiving your revised manuscript.

Kind regards,

Denis Alves Coelho, PhD

Academic Editor

PLOS ONE

Journal Requirements:

Additional Editor Comments:

Thanks for a thorough revision tackling all the reviewer comments. There are a few minor improvements suggested by one of the reviewers that we woud like you to consider in a new revision of your manuscript.

Reviewers' comments:

Reviewer's Responses to Questions

**Comments to the Author**

1. If the authors have adequately addressed your comments raised in a previous round of review and you feel that this manuscript is now acceptable for publication, you may indicate that here to bypass the “Comments to the Author” section, enter your conflict of interest statement in the “Confidential to Editor” section, and submit your "Accept" recommendation.

Reviewer #2: (No Response)

2. Is the manuscript technically sound, and do the data support the conclusions?

Reviewer #2: Partly

3. Has the statistical analysis been performed appropriately and rigorously? 

Reviewer #2: Yes

4. Have the authors made all data underlying the findings in their manuscript fully available?

Reviewer #2: Yes

5. Is the manuscript presented in an intelligible fashion and written in standard English?

Reviewer #2: No

6. Review Comments to the Author

Reviewer #2: The authors have taken a big step forward in revising this manuscript, but I still see some issues:

1. A small thing to start with: You use the word ‘compromises’ in two different places in the manuscript. I think you mean ’comprises’. (But on page 26, line 440, ‘compromising’ is used correctly.)

2. These sentences, page 8, lines 188-191, make me confused: “The presentation of the results compromises an overview of the study characteristics, and the quality assessed, a brief description of the interventions and their effectiveness. In line with the two research questions the results are structured into the presentation of interventions addressing occupational balance, and an appraisal of their effectiveness.” It seems to me that you present two ways of organizing the results. Besides, the first sentence is not grammatically correct. Based on how your results are actually presented, I suggest the following: “The presentation of the results comprises an overview of the study characteristics, a quality assessment, and a brief description of the interventions and their effectiveness. In line with the two research questions, the results are structured into the presentation of interventions addressing occupational balance, and an appraisal of their effectiveness.”

3. There is misuse of words like ‘hence’ and thus’. Delete as many as possible, but especially ‘thus’ on page 9, line 205, and substitute it with ‘and’. You already have ‘hence’ in the same sentence.

4. The number of participants in the BEL RCT is still not correct. It says very clearly in the paper from 2017 that there were 223 participants. I also think it is incorrect to view it as three RCTs if the papers are based on the same sample. It is quite common to write more than one paper from an RCT.

5. On page 21, lines 303 – 307, you have this long sentence: “The results indicated that both the experimental and control group, which showed no significant difference between the groups at baseline, improved on all outcomes measured from pre- to post-intervention and were therefore on par, with improvements in occupational balance scores (COPM and OBQ) being statistically significant (p ≤ 0.01) in both groups (57).” I am not sure I understand what you mean. I suggest you split the sentence into two and insert a dot after ‘intervention’. Then please rewrite the last part of the sentence for clarity.

6. The section starting on page 23, line 355, made me ponder a bit. I get the impression that social relations and digital solutions are something the authors are passionate about. It makes me think of argumentation rather than discussion of research findings. Okay, you give a reference to why social relations are important to occupational engagement (ref #60). But you gave no underpinnings to why a digital solution would be particularly suitable for an occupational balance intervention. Just your own thoughts and reasoning, which may be found true if tested, or perhaps not? I think you should nuance your text here. And perhaps move it down a bit and start with discussing what knowledge the articles have actually provided.

7. Page 24, line 399, you have the year (2018) instead of a number for the reference.

8. Please rewrite the sentence on page 26, lines 433-436, for clarity. Possibly a ‘which’ is missing on line 434.

9. Page 26, line 451: You write ‘arguable’. Do you mean ‘inarguable’?

10. Page 28, lines 487-491: Are you criticizing the studies you have reviewed for not being community-based, low-threshold services, or digital solutions? Or do you mean that existing research/interventions should be supplemented with such interventions? (I think many of the included interventions are in fact community based.) Again, I get the impression that you are more interested in conveying some messages than in discussing your findings. I think you could balance that a bit better. I absolutely agree with your arguments, as a citizen and human being; it is the academic researcher in me that protests.

11. Page 29, lines 512-513, you write that qualitative and mixed-methods research was not included in the literature review. Participatory research may fall under those categories. It seems a bit illogical to argue, as you indicate on page 28, lines 502-503, that participatory research is missing, since the reason may be that you have excluded that type of research. It is perhaps not likely, but you cannot know.

7. PLOS authors have the option to publish the peer review history of their article (what does this mean? ). If published, this will include your full peer review and any attached files.

**Do you want your identity to be public for this peer review?** For information about this choice, including consent withdrawal, please see our Privacy Policy .

Reviewer #2: No

---

## [Author Response · Author response to Decision Letter 2]

5 May 2025

Dear Reviewers and Dear Editor,

Thank you for your valuable feedback on our manuscript. We appreciate the time and effort you have taken to review our work. We have thoroughly addressed all your comments and believe that your input has again significantly improved the manuscript. Below, we have responded to each of your comments in detail.

1. Comment: A small thing to start with: You use the word ‘compromises’ in two different places in the manuscript. I think you mean ’comprises’. (But on page 26, line 440, ‘compromising’ is used correctly.)

Response: Thank you for making us aware of the incorrect use of the word. We have corrected it accordingly.

2. Comment: These sentences, page 8, lines 188-191, make me confused: “The presentation of the results compromises an overview of the study characteristics, and the quality assessed, a brief description of the interventions and their effectiveness. In line with the two research questions the results are structured into the presentation of interventions addressing occupational balance, and an appraisal of their effectiveness.” It seems to me that you present two ways of organizing the results. Besides, the first sentence is not grammatically correct. Based on how your results are actually presented, I suggest the following: “The presentation of the results comprises an overview of the study characteristics, a quality assessment, and a brief description of the interventions and their effectiveness. In line with the two research questions, the results are structured into the presentation of interventions addressing occupational balance, and an appraisal of their effectiveness.”

Response: We appreciate your feedback and have revised the introductory sentence accordingly, page 8, line 190-193: “The results section provides an overview of the study characteristics and a quality assessment of included studies. In line with the two research questions, it further comprises a presentation of interventions addressing occupational balance, and an appraisal of their effectiveness.”

3. Comment: There is misuse of words like ‘hence’ and thus’. Delete as many as possible, but especially ‘thus’ on page 9, line 205, and substitute it with ‘and’. You already have ‘hence’ in the same sentence.

Response: Thank you for pointing out the misuse of these words. We have carefully reviewed each instance throughout the entire manuscript and made the necessary replacements in the following places: line 38: “therefore”, line 101: “to date”, line 110: “therefore”, table 3: “thereby”.

4. Comment: The number of participants in the BEL RCT is still not correct. It says very clearly in the paper from 2017 that there were 223 participants. I also think it is incorrect to view it as three RCTs if the papers are based on the same sample. It is quite common to write more than one paper from an RCT.

Response: Thank you for your comment, which suggests that we have not yet described this clearly. As different authors report sample sizes at different time points in the context of BEL (and other included studies), we have tried to be as consistent as possible in this review. Accordingly, we initially decided to report the sample size of all studies based on the endpoint (i.e., the number of participants actually analyzed). While we had stated this in Table 1 (notes) in the previous version of the manuscript, your feedback highlights the need to be more explicit about reporting sample sizes in this review. We have now decided to report both the initial sample size at baseline (N) and the final number of cases analyzed (n analyzed) at the endpoint of studies. We have now clarified the rationale for this approach in several sections of the manuscript:

- In the method section, we describe the reasoning for presenting both sample sizes, lines 178-181: “Table 1 presents both the sample size at baseline (N) and the final sample size (n analyzed), which is the number of participants included in the analyses after accounting for dropouts. This ensures that the review is based on the actual number of participants who completed all relevant measurements (53).”

- Again, we report both sample sizes when we mention the number of participants in the 'Study characteristics' section, lines 215-218: “The sample size at baseline (N) varied between 12 (67) and 226 (37) participants across all studies, involving diverse populations, which are described in detail below. A total of 641 study participants completed all study measurements and were included in the analyses at the study endpoints (i.e., n analyzed).”

- In the description of sample sizes in Table 1 we now report both the initial sample size at baseline (N) and the final sample size in terms of the total number of participants analyzed (n analyzed), see column 5 of table 1, pp. 11-14.

- We have subsequently removed the number of trial participants from the abstract (line 16) so as not to mislead readers without additional information.

- When describing research on BEL and TTM starting from line 289ff, we are more specific about the sample sizes. BEL: In line 290 we have indicated the sample size at the start of the study. TTM: we state the sample size at baseline in line 304.

We trust that our approach is now presented in a clear and comprehensible manner.

In response to your valid observation that it is methodologically inaccurate to treat the studies as three distinct RCTs when they are based on the same sample, we have implemented the following modifications to enhance clarity.

- To provide further precision, we added the phrase 'of the included publications' (line 213) in the list of study designs, ensuring that the total number provided aligns with the number of included publications.

- For references that utilize data from the same study population as other cited references (for both BEL and TTM), Table 1 (pp. 11-14) includes notes indicating, “Analyses from the same study cohort as in (37) OR (58)” or “The study cohort is part of the cohort in (37)”

- Under the heading “the effectiveness of interventions targeting occupational balance” (starting on page 20, line 287), we have provided a more detailed explanation indicating that these are distinct publications derived from a single RCT study (for both BEL and TTM).

5. Comment: On page 21, lines 303 – 307, you have this long sentence: “The results indicated that both the experimental and control group, which showed no significant difference between the groups at baseline, improved on all outcomes measured from pre- to post-intervention and were therefore on par, with improvements in occupational balance scores (COPM and OBQ) being statistically significant (p ≤ 0.01) in both groups (57).” I am not sure I understand what you mean. I suggest you split the sentence into two and insert a dot after ‘intervention’. Then please rewrite the last part of the sentence for clarity.

Response: Thank you for highlighting this issue. In response to your suggestion, we have revised the sentence by splitting it and rephrasing the latter part for enhanced clarity, as follows: (lines 307-311): “The results of the original RCT conducted in 2018 indicated that both the experimental and control group improved on all outcomes measured from pre- to post-intervention. The improvements in occupational balance scores (COPM and OBQ) were statistically significant (p ≤ 0.01) in both groups, indicating that the intervention was not significantly better than regular occupational therapy (58).”

6. Comment: The section starting on page 23, line 355, made me ponder a bit. I get the impression that social relations and digital solutions are something the authors are passionate about. It makes me think of argumentation rather than discussion of research findings. Okay, you give a reference to why social relations are important to occupational engagement (ref #60). But you gave no underpinnings to why a digital solution would be particularly suitable for an occupational balance intervention. Just your own thoughts and reasoning, which may be found true if tested, or perhaps not? I think you should nuance your text here. And perhaps move it down a bit and start with discussing what knowledge the articles have actually provided.

Response: Thank you for this valuable comment. We appreciate your insight and have revised the discussion section accordingly. We have made efforts to formulate the thematic emphasis on digital solutions and social relations (see lines 369 - 374, lines 410 - 412). To soften the emphasis on social relations, we have moved this section further back and rewritten it to present a more nuanced perspective.

7. Comment: Page 24, line 399, you have the year (2018) instead of a number for the reference.

Response: Thank you for pointing this out. We have now correctly cited the sentence (line 388). Additionally, we have carefully reviewed the entire manuscript and corrected similar instances in lines 61, 66, 205, 294, 298, and 312.

8. Comment: Please rewrite the sentence on page 26, lines 433-436, for clarity. Possibly a ‘which’ is missing on line 434.

Response: Thank you for this comment. We have reformulated the sentence for clarity (see lines 426 - 429): “As another example, two studies that reported beneficial effects for the AOI intervention and the ballroom dancing intervention, were rated as low quality due to small sample sizes, significant dropout rates and the use of a non-validated outcome measure.”

9. Comment: Page 26, line 451: You write ‘arguable’. Do you mean ‘inarguable’?

Response: Thank you for this comment. We implemented your suggestion (line 443).

10. Comment: Page 28, lines 487-491: Are you criticizing the studies you have reviewed for not being community-based, low-threshold services, or digital solutions? Or do you mean that existing research/interventions should be supplemented with such interventions? (I think many of the included interventions are in fact community based.) Again, I get the impression that you are more interested in conveying some messages than in discussing your findings. I think you could balance that a bit better. I absolutely agree with your arguments, as a citizen and human being; it is the academic researcher in me that protests.

Response: We fully concur with your observation. Thank you for highlighting this important issue. We have revised the section, accordingly, lines 479 - 486: “Scholars agree that interventions addressing occupational balance may be promising from a public health perspective (13, 34). While some interventions followed a community-based approach, most of them have been explored in clinical in- or outpatient healthcare settings or were delivered by institutions outside the healthcare sector, e.g., universities. Future research should supplement existing interventions with more community-based, low-threshold services, or digital solutions. Expanding the scope to include more diverse and accessible formats will help in increasing their public health impact.”

11. Comment: Page 29, lines 512-513, you write that qualitative and mixed-methods research was not included in the literature review. Participatory research may fall under those categories. It seems a bit illogical to argue, as you indicate on page 28, lines 502-503, that participatory research is missing, since the reason may be that you have excluded that type of research. It is perhaps not likely, but you cannot know.

Response: We appreciate your observation, thank you. To clarify and avoid any perceived contradiction, we have reorganized the argumentation. We acknowledge that excluding participatory research is a limitation of our review. Consequently, we have linked these points and grouped them under the heading of limitations, explicitly stating that the absence of qualitative research in our review is considered a limitation, lines 501 - 515: “Due to the chosen methodology, interventions studied using qualitative or mixed-method designs were not included in this review. Acknowledging the subjectiveness of the occupational balance concept as well as the fact that it may differ across cultures, age groups and populations is crucial. Future projects aiming to design occupational balance interventions may follow participatory research approaches and truly involve the population of interest throughout the design process. In fact, additional interventions promoting occupational balance were detected, which still need to be examined for their effectiveness with quantitative methods. Among the interventions discovered were Project Bien Estar (100), self-management occupational therapy program (SMOoTh) (101), educational workshop on time use (102), inpatient energy management education (IEME) (103), mindful based program (104), home modification intervention (105), psychological rehabilitation program (106), therapeutic gardening (33) and Daily Life Coping (107). Given that these interventions demonstrate promising approaches and existing research may already incorporate the aforementioned participatory methods, further investigation is warranted.”

12. Comments to the authors: The manuscript ist not written in standard English / language issues

Response: The entire CROB research team and a native speaker have reviewed the manuscript linguistically and made improvements at several points, all of which have been marked in the manuscript for tracking.

13. Journal requirements: Please review your reference list to ensure that it is complete and correct. If you have cited papers that have been retracted, please include the rationale for doing so in the manuscript text, or remove these references and replace them with relevant current references. Any changes to the reference list should be mentioned in the rebuttal letter that accompanies your revised manuscript. If you need to cite a retracted article, indicate the article’s retracted status in the References list and also include a citation and full reference for the retraction notice.

Response: We have checked the reference list for retracted entries, and no references needed to be corrected.

We hope that our revisions align with your expectations and effectively address your concerns. Thank you once again for your insightful feedback.

Best regards,

Stefanie Lentner

---

## [Editor Report · Decision Letter 2]

6 May 2025

Interventions promoting Occupational Balance in Adults: A Systematic Literature Review

PONE-D-24-45446R2

Dear Dr. Lentner,

We’re pleased to inform you that your manuscript has been judged scientifically suitable for publication and will be formally accepted for publication once it meets all outstanding technical requirements.

Kind regards,

Denis Alves Coelho, PhD

Academic Editor

PLOS ONE

---

## [Editor Report · Acceptance letter]

PONE-D-24-45446R2

PLOS ONE

Dear Dr. Lentner,

I'm pleased to inform you that your manuscript has been deemed suitable for publication in PLOS ONE. Congratulations! Your manuscript is now being handed over to our production team.

Kind regards,

on behalf of

Dr. Denis Alves Coelho

Academic Editor

PLOS ONE